# Reduction of NAD and NMN on mineral surfaces with H₂ reveals a functional role for the AMP moiety in a prebiotic context
Delfina P. Henriques Pereira [1,2] ✉, Xiulan Xie[3], Sarah V. Stewart [4,5], Zainab Subrati[1,2], Tuğçe Beyazay[6], Nicole Paczia [7], Jürgen Belz[8], Kerstin Volz[8], Valentina Erastova[4,5], Harun Tüysüz[6,9] & Martina Preiner [1,2] ✉

Many cofactors share a molecular structure – adenosine monophosphate (AMP) – that otherwise occurs in nucleic acids. The presence of AMP in cofactors has presented an evolutionary puzzle. Is it a biochemical 'handle' that allows proteins to bind the cofactor more tightly, or a relic from a prebiotic time when cofactors arose? Using the example of nicotinamide adenine dinucleotide (NAD), we find a previously unknown property of its AMP handle. Experiments with hydrogen gas on mineral surfaces show that the handle-free nicotinamide mononucleotide (NMN) overreduces quickly, while NAD gets reduced specifically. The handle allows NAD to function in a hydrothermal, mineral-based setting, indicating that it is a form of protection against a harsh environment. Our findings uncover a specific functional role for the AMP moiety of NAD under environmental conditions capable of non-enzymatic NAD reduction, thereby identifying a structural element of a redox cofactor that is older than the enzymes using it.

In many theories for the origin of metabolism, organic cofactors such as $NAD^+$, flavin adenine dinucleotide (FAD), S-adenosyl methionine (SAM), and coenzyme A (CoA) are seen as crucial intermediates in the transition from inorganic to enzymatic catalysis[1–7]. These cofactors catalyze ancient biochemical reactions with a wide variety of substrates, are often active in the absence of enzymes[8–12], and share a 'handle' consisting of the adenosine nucleotide (nucleoside in the case of SAM), a central component of nucleic acids such as RNA.

The catalytic and nucleotide moieties of cofactors like $NAD^+$ thus unite "metabolism first" and "information first" views for the origin of life[1,7,13] in a single molecule (Fig. 1). In all enzymatic reactions known to date, the adenosine moiety is catalytically inert, in non-enzymatic reactions, its phosphate groups have been shown to enable association to metal ions[9,11].

Some cofactors that do not possess the moiety, such as pterins or folates, nonetheless derive from nucleotides in their biosynthesis[5,14–16], indicating close connections between cofactor catalysis and RNA bases in early biochemical evolution. The evolutionary rationale behind the conserved presence of an adenosine moiety to $NAD^+$ is traditionally viewed either functionally, in terms of a biochemical handle that allows enzymes to

bind the cofactor more efficiently[17,18], or historically also as a holdover from an earlier phase of evolution in which RNA preceded enzymes and cofactors were active in ribozymes[1]. But there is, however, a third possibility. $NAD^+$ functions exclusively as a redox cofactor, and neither traditional view takes into account the environmental source of electrons required to generate NADH or reduced substrates for $NAD^+$ dependent reactions.

NAD is an ancient redox cofactor that is essential in metabolism and traces back to the last universal common ancestor, LUCA[2,19–21]. Several prebiotic routes of NAD synthesis have been proposed, including via mineral-assisted synthesis under hydrothermal conditions[6,22] or via nitrile-dependent syntheses[13,23,24]. Recent work has shown how $NAD^+$ can be non-enzymatically reduced under a variety of conditions[8,9,12,25,26]. Additionally, the reducing abilities of NADH have been demonstrated both with and without metal ions as catalysts[11,27].

What is a good possible primary electron source and thus geochemical setting? In this study, we are investigating the abilities of water-rock-interaction systems, where protons of water are being reduced to hydrogen ($H_2$) gas by electrons of Fe(II) containing minerals (serpentinizing systems). Previous studies have shown that $NAD^+$ readily reacts

[1]Microcosm Earth Center, Max-Planck-Institute for Terrestrial Microbiology and Philipps-University Marburg, Marburg, Germany. [2]Geochemical Protoenzymes Research Group, Max-Planck-Institute for Terrestrial Microbiology, Marburg, Germany. [3]Department of Chemistry, Philipps University Marburg, Marburg, Germany. [4]School of Chemistry, University of Edinburgh, Edinburgh, United Kingdom. [5]UK Centre for Astrobiology, School of Physics and Astronomy, University of Edinburgh, Edinburgh, United Kingdom. [6]Heterogeneous Catalysis, Max-Planck-Institut für Kohlenforschung, Mülheim an der Ruhr, Germany. [7]Metabolomics and small molecule mass spectrometry, Max-Planck-Institute for Terrestrial Microbiology, Marburg, Germany. [8]Department of Physics, Philipps University Marburg, Marburg, Germany. [9]IMDEA Materials Institute, Madrid, Spain. ✉e-mail: delfina.pereira@mpi-marburg.mpg.de; martina.preiner@mpi-marburg.mpg.de

**Fig. 1 | A selection of central organic cofactors.**
These cofactors display an adenosine-based handle
(red) connected to a functional part that determines
the role of these cofactors in metabolism (black):
Electron/hydride transfer, methyl transfer or acetyl-
transfer. In purple, the function-associated half of
nicotinamide dinucleotide (NAD) is highlighted:
nicotinamide mononucleotide (NMN).

with $H_2$ and metal powders (Ni, Co, Fe) to specifically form the biolo-
gically relevant form of reduced NAD (1,4-NADH)[12]. An independent
study also reported $NAD^+$ reduction without $H_2$, using iron sulfides as
reductant at lower yields[25].

The conditions of serpentinizing systems not only reduce $NAD^+$ but
also carbon dioxide ($CO_2$) with the electrons provided by $H_2$[20,21,28,29]. They
are rich in Fe and, depending on the system, Ni as well[30], just like enzymes of
the acetyl CoA pathway, which $H_2$-dependent anaerobic autotrophs use to
reduce $CO_2$ in their carbon and energy metabolism[31,32]. Hotter systems tend
to feature Ni-Fe alloys with higher nickel composition, while cooler ones are
richer in iron[30]. Nickel-containing intermetallic compounds such as awar-
uite ($Ni_3Fe$) or taenite ($NiFe_3$ to $Ni_2Fe$) are products of the reaction of $H_2$
with Ni(II) compounds in serpentinizing systems[30,33,34] and are also found in
meteorites[35]. Native metals occur naturally in these highly reducing
systems[36,37]. Here, we investigate the ability of naturally occurring Ni- and
Fe-containing alloys to replace enzymes for $H_2$-dependent $NAD^+$ reduc-
tion. We then compare NAD to its AMP-lacking homolog NMN to test how
the AMP moiety impacts the nature of products obtained from non-
enzymatic (prebiotic) $NAD^+$ reduction with $H_2$.

## Results

### Screening naturally occurring iron and nickel-containing minerals

Ni-Fe containing minerals found in hydrothermal settings were tested for
the reduction of $NAD^+$ under conditions comparable to those found in mild

serpentinizing hydrothermal systems (40 °C, 0.133 M phosphate buffer pH
8.5, 5 bar $H_2$, Supplementary Scheme S1). These nanoparticular mineral
powders were synthesized via the nano-casting method by using tea leaves
as a template and were previously characterized[38,39]. The metal content in
these reactions is equivalent to that of the cofactor (1 metal atom per
cofactor). The resulting $H_2$ concentration at our conditions is 3.6 mM (using
Henry's law, s. Supplementary Equations S1–3, Wimmer et al.[19] and
Schwander et al.[40]), which is comparable to the $H_2$ concentrations found in
the effluent of serpentinizing systems[41,42]. The buffer was bubbled with $N_2$
for 1 h and handled inside a glove box to approximate the anoxic conditions
on early Earth. Several controls were implemented, including controls
without metal and $H_2$, respectively. The liquid phase was analyzed
by $^1H$-NMR.

After 4 h under $H_2$, 1,4- and 1,6-NADH formed in all samples at
different yields, and the reaction with nanoparticular $NiFe_3$ ($nNiFe_3$)
yielding the most of both molecules (Fig. 2). Control experiments starting
from 100% 1,4-NADH, showed that 1,6-NADH is a product of rearran-
gement from 1,4-NADH that occurs spontaneously without the need of a
catalyst (Supplementary Scheme S2, Table S4, Fig. S3). Starting from $NAD^+$,
the proportion of 1,6-NADH is higher in samples with higher NADH yields
(Ni-Fe alloys), showing that the accumulation of 1,6-NADH is not entirely
independent of the metal (Supplementary Table S5).

Samples under Ar also produced NADH with Fe-rich minerals
($nNiFe_3$, $nFe$). In addition to transferring hydrides from $H_2$ to $NAD^+$, iron
can oxidize, donating its own electrons either by producing nascent $H_2$ gas,

**Fig. 2 | NAD+ reduction at 40 °C over 4 h.** Equimolar amounts (normalized to 1 metal atom per NAD molecule) of several Ni-Fe alloys are used under 5 bar of $H_2$ (or Ar). **A** Segment of the $^1H$-NMR spectra where the chemical shift of the hydrogen on the second nicotinamide carbon is visible upon reduction. 1,4-NADH features a characteristic peak at $\delta = 6.9$ ppm and 1,6-NADH at $\delta = 7.1$ ppm (Supplementary Table S1). **B** Yield of 1,4-NADH obtained for several metals after 4 h under $H_2$ and Ar. Reduction under Ar is detected only with minerals whose metal content is $\geq 75\%$ Fe. With 5 bar of $H_2$, all metals can facilitate 1,4-NADH synthesis. Mixed alloys are more efficient than pure metals. ND means not detected, error bars show standard deviation (SD), and hollow circles show individual measurement points. All spectra and yields can be found in Supplementary Tables S2–3, Figs. S1 and S2.

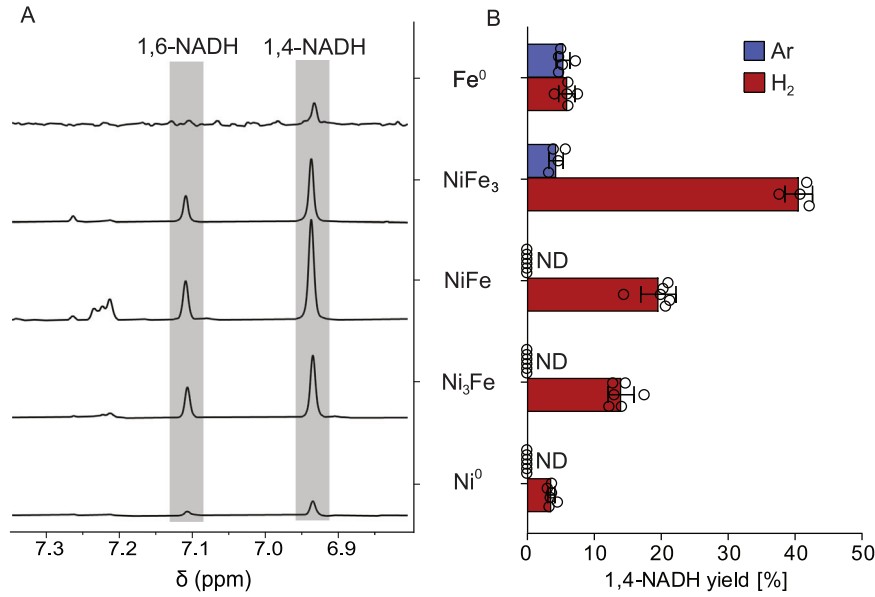

ultimately reducing NAD+ or by direct electron transfer to NAD+. This process can also be used as a proxy for the constant $H_2$-production in serpentinizing systems[12]. Ni by itself is $H_2$-dependent in the promotion of NAD+ reduction.

Scanning transmission electron microscopy (STEM) imaging before and after the reactions (the latter including a washing and dilution step to assure true surface alteration) confirms that Fe, both under Ar and $H_2$, gets associated with phosphate ions in a ratio that suggests the formation of iron phosphate. Ni does not associate with phosphate, suggesting it stays in its native form (Supplementary Methods and Figs. S4–7).

In Ni-rich minerals, the Fe is expected to slowly reduce NAD under Ar conditions. However, it is likely that the resulting products do not reach the detection threshold within 4 h. Overall, bimetallic minerals are significantly more efficient than the individual transition metals when hydrogen is available. Introducing one Ni atom to a Fe atom increases the yield by 300% (nNiFe vs. nFe). Their properties, already observed in a previous study[12], seem to complement each other for the reduction of NAD+ with $H_2$: Fe being mostly an electron donor, while Ni promotes hydride transfer from $H_2$. These complementary roles have also been described in other publications, suggesting charge transfers from Fe (more electropositive) to Ni could increase the electron density in Ni[38,43].

### The "universal" adenine nucleotide in organic cofactors

Many central cofactors share an AMP "handle" (Fig. 1) attached to the catalytically active moiety. In the case of NAD, NMN is the hydride-transferring nicotinamide; AMP is inert. NAD is stable in water, with its pH range depending on the reduction state of the nicotinamide: NADH is more stable at pH > 7, while NAD+ is more stable under acidic conditions[12]. To investigate the role of the AMP-tail in a prebiotic context, several experiments were designed to compare NAD and NMN. We initially focused on nNiFe3, the most efficient of the Ni-Fe minerals in the above-described NAD experiment (Fig. 2). All other reaction conditions (buffer, pH, temperature, metal to cofactor ratio) were maintained (Supplementary Scheme S3). Products were quantified via $^1H$-NMR spectroscopy.

Under Ar, NMN got reduced due to the iron in the mineral compound (nNiFe3) working as an electron donor, but more slowly than under $H_2$, where Ni can function as a hydrogenation catalyst (Supplementary Tables S6 and S7). Without metals, NMN does not react and remains stable, regardless of the gas phase (Supplementary Figs. S8 and S9).

In addition to the 4 h experiment above, a 2 h experiment with NAD+ under $H_2$ showed the increase of 1,4- and 1,6-NADH to be steady and inversely proportional to the decrease of NAD+ in solution (Fig. 3D). After 4 h with nNiFe3, on average 57% of NAD+ was reduced with 26% remaining oxidized. The remaining 17% can in part be attributed to degradation to nicotinamide, but also unassigned degradation reactions and loss via surface absorption[44].

NMN, however, shows a completely different reaction profile (Fig. 3C). After only 1 h, 69% of the starting NMN had been consumed, and a variety of products were observed in 1D $^1H$-NMR (Fig. 3A). 2D-NMR spectroscopy facilitated the identification of the overreduction of NMN's nicotinamide ring with two and three hydrogenation sites, so 1,4,6-products (**2c**), and a 1,2,4,6-product (**2d**) respectively (Fig. 3A, B and D). While the fully reduced species **2d** formed early and its concentration does not change significantly over time and stays below 3%, the concentration of twice reduced products **2c** increases steadily over time and correlates with a decrease of 1,4-NMNH. This indicates that not all reductions might be step-wise processes (Suppl. Fig. 3B), especially in the case of **2d**.

Under Ar, **2d** did not form at all with nNiFe3, demonstrating that $H_2$ is necessary for the full hydrogenation of the nicotinamide ring (Supplementary Table S7 and Fig. S9). **2c**, however, also formed under Ar, albeit in far lower yields (3%) than under $H_2$ (25%) after 4 h. The yield of 1,4-NMNH was relatively similar in both atmospheres (7% under Ar; 10% under $H_2$). Transferring these observations to environmental conditions suggests that less reducing conditions could be favorable for specific NMN reduction.

After 1 h under $H_2$, 1,4-NMNH was the main product (Fig. 3A and D). Other side products formed at a comparable rate, rapidly depleting the reagent NMN. Consequently, the production of 1,4-NMNH seems to have stopped after 2 h and subsequently began to decrease in concentration. The concentration of **2c** continuously increased over time. Even though the concentration of 1,4-NMNH decreased from 35% to 9% in 2 h, the total amount of reduced NMN remained relatively stable, between 67 and 69% (Fig. 3C). This suggests that 1,4-NMNH is the first and main product of NMN reduction, which can subsequently undergo further reduction to other species, mainly **2c**.

We were able to exclude two products commonly found in NAD+ reduction, where C2 or C6 of the nicotinamide ring is reduced[26,45,46]. Reduction products presumably starting with these two one-time reduced products could be excluded (Supplementary Figs. S26–29 and Scheme S7). In the case of NADH, its 1,2-reduced form is known to be unstable, so it is likely this is the case with 1,2-NMNH as well, leading to its absence in our reaction[47].

In the case of 1,4-NMNH loss over time, several routes exist: (i) mainly the further reduction to **2c**, (ii) 1,4-NMNH becoming hydrolyzed at C5 or

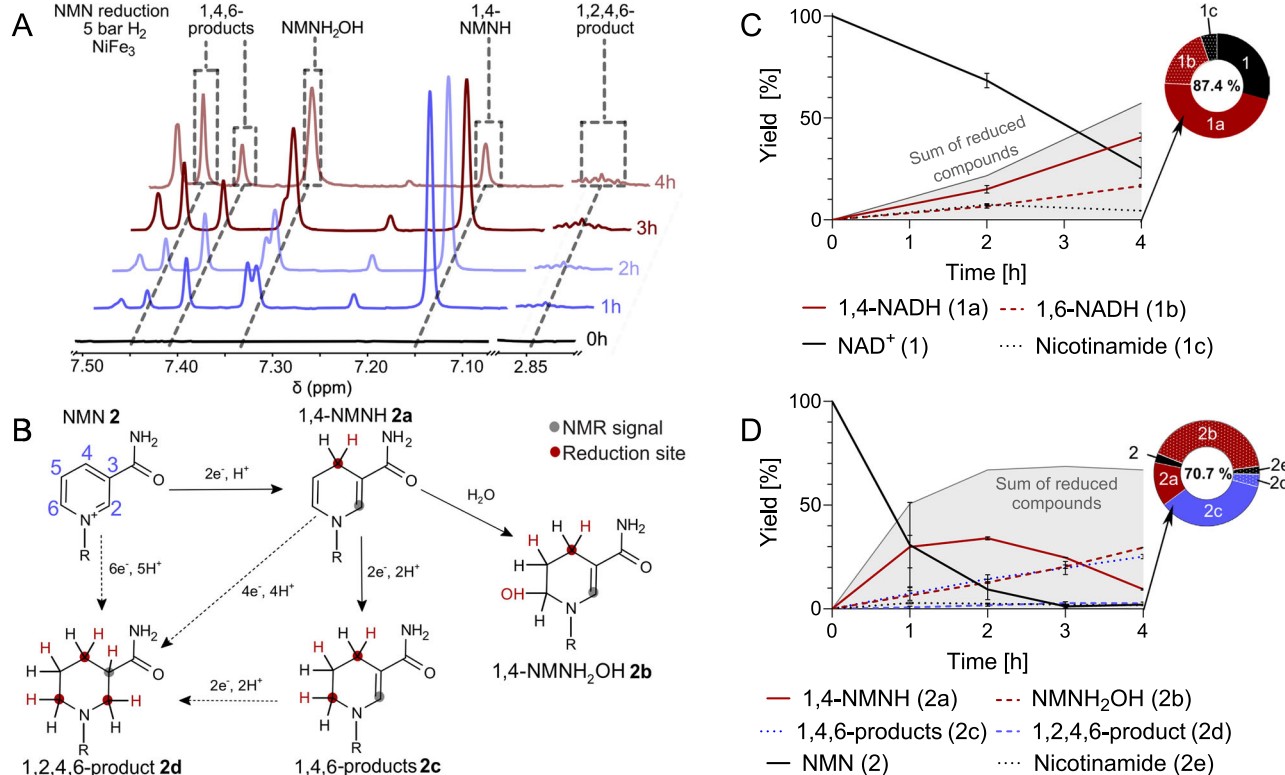

**Fig. 3 | The reduction of NMN and NAD⁺.** This reaction, promoted with equimolar quantities of NiFe₃ nanopowder under 5 bar of H₂ and at 40 °C, was monitored for 4 h. The nanopowder was normalized to the number of metal atoms. Both reactions were carried out in oxygen-free, aqueous solutions at a pH of 8.5 (0.133 M phosphate buffer). **A** ¹H-NMR (600 MHz) spectra monitoring NMN reduction over time. Dashed lines mark the peaks used for identifying and quantifying reduced NMN species previously identified via 2D NMR (Supplementary Table S8). Due to the complexity of the mixture, not all peaks could be assigned (Supplementary Figs. S10–19 and Scheme S4). **B** Proposed reduction patterns for NMN reduction with H₂ and Ni-Fe minerals. The reduction site and additional protons are highlighted in red. Full arrows represent proposed reactions supported by the data obtained, while dashed arrows are reactions that could not be entirely confirmed or excluded. Gray circles indicate the proton providing the NMR signal for

C6, and (iii) 1,4-NMNH engaging in various dimerization reactions with 1,6-NMNH (Diels–Alder type reactions; Supplementary Scheme S8). Via liquid chromatography mass spectroscopy (LC–MS), we were able to exclude such products and confirmed the presence of a hydration product, so an OH⁻ being added to 1,4-NMNH (Fig. 4B, Supplementary Figs. S30–32). The hydration product NMNH₂OH (**2b**) was matched to a peak at 7.34 ppm in the ¹H-NMR (Supplementary Figs. S18 and S19); the adjacent smaller, overlapping peak at 7.35 ppm could not be assigned beyond a doubt area of the smaller peak was thus subtracted with a deconvolution tool (Supplementary Table S10–12). The two peaks assigned **2c** have been attributed to twice-reduced nicotinamide rings as depicted in Fig. 3B. This is either due to conformational isomers or possibly other deviations of the molecule apart from the nicotinamide ring.

An often reported side-product of NAD reduction (e.g., via cyclic voltammetry) is a 4,4'-linked NAD dimer[48], which also qualifies as a possible side reaction of NMN reduction. Here, after careful interpretation of our 2D NMRs of the 1 h and 4 h reaction with NMN and the 4 h reaction with NAD, we can exclude the presence of such dimers (Supplementary Fig. S13; no peak at 40–50 ppm in ¹³C of a bridging methine corresponding to the linkage). This was also confirmed via LC–MS (no double charged molecules were detected). As these dimers are a direct result of radical-forming 1e⁻ transfers onto NAD[49], we can draw the conclusion that direct hydride or 2e⁻ transfer is the present mechanism in our reactions.

quantification. In bold font, numeral assignments for NMN reduction products are made. **C** Time course of NAD⁺ reduction (Supplementary Scheme S5). Reduced NAD species are plotted as relative percentage to a metal-free control sample (Supplementary Methods)—all time points represent the mean and SD of the same reaction (2 h: n = 2, 4 h: n = 4). The gray area shows the sum of all reduced compounds (**1a** and **b**). **D** Time course of NMN reduction (Supplementary Scheme S6). Reduced NMN species are plotted as relative percentage to a metal-free control sample. All time points represent the mean and SD (1 h and 3 h: n = 3; 2 h and 4 h: n = 2) of the same reaction. The ring chart represents the distribution of products after 4 h, percentage in the center indicates the entirety of assigned products. The gray area shows the sum of all reduced products (**2a–d**). Yields shown in **C** and **D** are also listed in detail in Supplementary Tables S9–13, Supplementary Figs. S20–25.

After quantification of all identified species, we can account for at least 70.7% of transformed NMN for all reactions, often more. Unidentified species encountered in lower yields can also stem from the differently reduced versions of the degradation product nicotinamide[50]. It is furthermore possible, as mentioned above, that some NMN was lost due to interaction with the mineral surface. Overall, there is a notable and surprising difference between the reduction profile of the dinucleotide and the mononucleotide.

In order to evaluate these differences further, we performed several molecular dynamics calculations (Supplementary Methods, Supplementary Tables S15 and S16 and Figs. S34–44). NAD is known to dynamically fold in aqueous solution[51–55]. Here, we observe that although NAD interchanges between folded and unfolded conformation in solution, the alternation mostly stops on the metal surface (the simulations were performed on a Ni surface). On the surface, NAD⁺ occurs in stabilized open and closed configurations (Supplementary Figs. S35b and S36), with 30–40% being folded. Once reduced to 1,4-NADH, the adsorption to the surfaces decreases (Supplementary Fig. S43, Supplementary Table S16).

**Different metals, different mechanisms**
Starting from the observation that during NMN reduction, 1,4-NMNH is a main product decreasing with the length of the reaction, we hypothesized that less efficient catalysts might help to avoid overreduction and thus

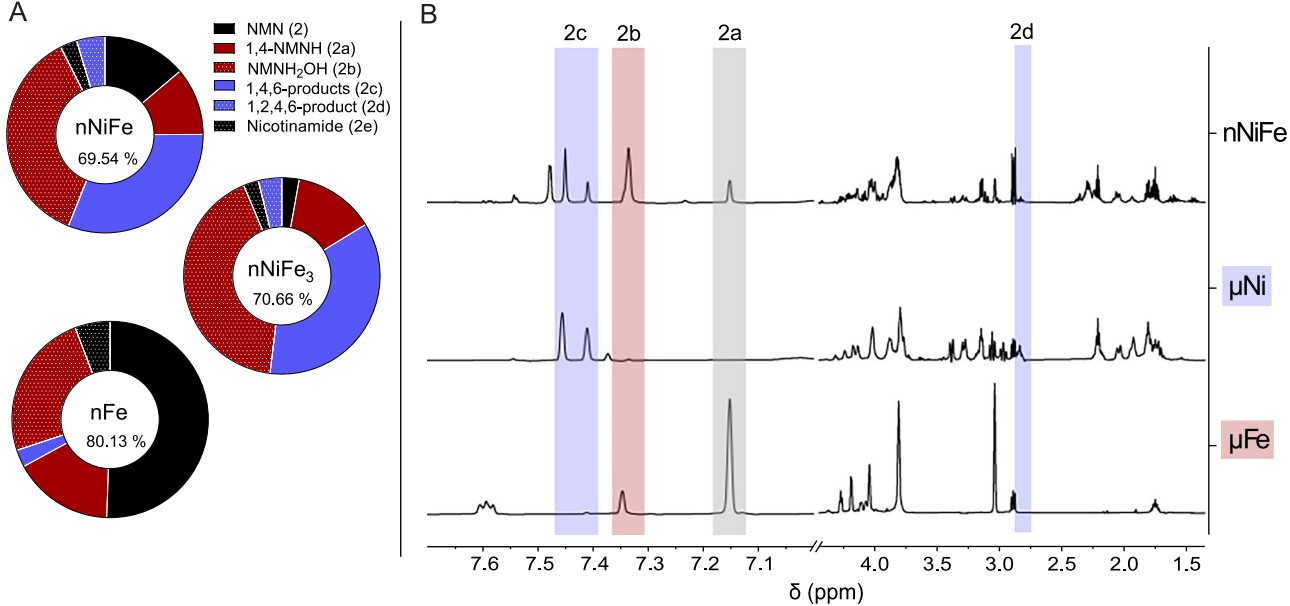

**Fig. 4 | Selectivity and efficiency of NMN reduction with $H_2$ gas are influenced by the type of metal prevalent. A** Product spectrum of NMN reduction over 4 h (5 bar, 40 °C) with decreasing Ni ratio. The proportion of overreduction products decreases abruptly when no Ni is used (nFe), but so does the overall NMN conversion (Supplementary Table S6 and Fig. S8). **B** The reduction of NMN assisted by nNiFe after 4 h (cofactor ratio 1:1) yields products that directly correlate with the products obtained when utilizing the metals separately in micropowder form (metal-cofactor ratio 200:1). $\mu Ni^0$ promotes the accumulation of **2c** and **2d** (purple) and other side products. 1,4-NMNH (gray) seems not to react further when using $\mu Fe^0$, accumulating over time, but hydrolyzes in parts further to **2b**. Note that this comparison is a qualitative one, as it would not be appropriate to compare the yields between nano- and micropowder. Yields for micropowder reactions shown in this figure are listed in Supplementary Table S14 and Fig. S33.

reduce NMN more specifically than the previously used rather efficient nanoparticular Ni-Fe alloys. As both nanoparticular Ni and Fe (nNi and nFe) visibly worked less efficiently for $NAD^+$ reduction, but Fe promoted reduction in higher yields, we decided to work with the same powder (nNiFe, nFe; 1:1 ratio to the cofactor) used for $NAD^+$ reduction (Fig. 4A). Reducing the ratio of Fe (nNiFe) also reduces the amount of converted NMN, while keeping the ratio of 1,4-NMNH to side products quite similar. The nFe powder by itself converted far less NMN but also did not promote the formation of three times reduced **2d**, while twice reduced **2c** is only produced in almost untraceable amounts. The hydration product **2b** is visible as a peak at 7.35 ppm, still accumulated over the 4 h reaction time.

We repeated experiments with NMN and $H_2$ over commercially available Fe and Ni micropowder ($\mu$Fe, particle size: <150 $\mu$m; $\mu$Ni, particle size: 3–7 $\mu$m). This separates the metal dependency from the general reduction efficiency of nanoparticular powders due to a large surface area. The metal-cofactor ratio was 200:1 to guarantee the detection of even low concentration side products. The results (Fig. 4B) show a remarkable trend to overreduction with $\mu$Ni, while $\mu$Fe mostly displays two main products: 1,4-NMNH and **2b**, the latter being the hydrolysis product of the former, which forms with and without a metal catalyst (Supplementary Scheme S9, Table S17, and Fig. S45). Comparing the spectra of NMN reduction with $\mu$Ni only and $\mu$Fe only with those of nNiFe, the distinct product patterns of each metal become apparent.

Nickel has long been recognized as a hydrogenation catalyst[56]. But why does it, when not combined with Fe, only reluctantly reduce NAD (Fig. 2 and Pereira et al.[12]) and yet overreduce NMN, not leaving any traceable amount of single-reduced species? The answer, we suggest, lies again in the structural differences between NAD and NMN. NMN can be more easily absorbed to a hydrogenated Ni surface, possibly over the whole nicotinamide ring (Supplementary Scheme S10). This could also explain the fast formation of its fully reduced product **2d** shown in Fig. 4C. NAD in a staggered formation could only absorb partly on the surface, avoiding such overreduction reactions[57]. Molecular dynamics simulations on a Ni surface

show that once NMN is reduced to 1,4-NMNH it tends to stick around at the surface more than NAD after reduction (Supplementary Figs. S42, Supplementary Table S15). If so, why does Fe not overreduce NMN as readily as Ni? Here, we can reflect on the mechanisms postulated by us in Pereira et al.[12], that Fe both serves as a (less effective) hydrogenation catalyst and a strong electron donor, either via direct electron transfer to the nicotinamide cofactor or the formation of nascent hydride groups on its surface. Assuming that Fe predominantly reduces NMN through direct electron transfer, the reduction process prioritizes the species with the most favorable redox potential first: 1,4-NMNH (and 1,4-NADH, in the case of NAD). This hypothesis was substantiated by cyclic voltammetry (CV) measurements, which revealed that 1,4-NMNH exhibits the lowest reduction potential among all the reduction products obtained from NMN (Supplementary Table S18, Figs. S46 and S47), meaning it is the first to get oxidized. Another possible explanation could be that the Ni catalyst does not alter as much as the Fe surface, meaning there would be a constant supply of hydrides available. For Fe, the previously described association with phosphate from the buffer could block active centers, which further prevent overreduction. To investigate this further, we performed experiments with $\mu$Fe and NMN under $H_2$ in carbonate buffer (Supplementary Schemes S11 and S12), leading to the same yields as phosphate buffer (Supplementary Schemes S11–14, Supplementary Tables S19–23, and Figs. S48–52). This indicates that oxidation of Fe and precipitation with the buffer's anions can be a reason for the different outcome of Fe and Ni reactions, but at the same time excludes a catalytic effect of phosphate in the reaction[58]. While the combination of nickel's hydrogenation strengths and iron's electron donation increases the yield of 1,4-NADH immensely compared to Fe or Ni separately (Fig. 2), the addition of Ni does not increase the directed reduction of NMN to 1,4-NMNH (Fig. 4A). The lacking overreduction with Fe also explains the accumulation of the hydrolysis product **2b** in Fe-only reactions: if more 1,4-NMNH can be formed without being further reduced, the more of it can be hydrolyzed to **2b**. So with a strong hydrogenation catalyst such as Ni, overreduction likely prevents hydrolysis.

**Table 1 | Overview of yields of mixtures of NAD$^+$ and NMN in comparison to separate reduction with H$_2$ gas**

| | Individual (12 mM) | | | | Competition (12 mM each) | | | |
|---|---|---|---|---|---|---|---|---|
| | Fe | | Ni | | Fe | | Ni | |
| **1,4-NADH (1a)** | **8.12%** | ±1.25 | **19.89%** | ±0.15 | **11.33%*** | ±1.20 | **13.63%*** | ±0.83 |
| 1,6-NADH (1b) | 1.60% | ±0.24 | 5.26% | ±0.15 | 2.17% | ±0.28 | 3.51% | ±0.35 |
| NAD$^+$ | 57.52% | ±2.85 | 53.12% | ±1.82 | 45.10 | ±10.60 | 63.82% | ±17.08 |
| **1,4-NMNH (2a)** | **47.95%** | ±11.89 | **4.85%** | ±3.47 | **7.06%*** | ±0.93 | **5.70%*** | ±0.92 |
| NMNH$_2$OH (2b) | 11.43% | ±2.03 | 13.79% | ±4.53 | 2.26% | ±0.71 | 7.33% | ±0.31 |
| 1,4,6-products (2c) | 0.00% | ±0.00 | 44.13% | ±7.07 | 0.00% | ±0.00 | 3.74% | ±0.95 |
| 1,2,4,6-product (2d) | 0.00% | ±0.00 | 9.83% | ±3.11 | 0.00% | ±0.00 | 0.00% | ±0.00 |
| NMN | 23.48% | ±15.84 | 0.92% | ±0.40 | 75.15% | ±11.26 | 79.55% | ±1.75 |

The left column shows the quantification of 12 mM of NAD$^+$ ($n = 3$) and 12 mM of NMN ($n = 3$) in individual reactions with µNi and µFe. The right column shows a reaction mixture of the same amount of NAD$^+$ and NMN (12 mM each) (for all $n = 3$). In all experiments, the metal powder concentration lies at 600 mM, so for individual reactions, the metal to cofactor ratio is 50:1, for competition reactions it is 25:1. All yields are calculated per to 12 mM of each starting cofactor, all SD values shown next to the yield values. Unpaired t-tests were used to evaluate whether the differences in concentration between 1,4-NADH and 1,4-NMNH (bold values) in the competition experiments are significant: *two-tailed $P$ value = 0.0166, significant difference; ***two-tailed $P$ value = 0.0008, very significant difference. All additional data for these experiments can be found in Supplementary Tables S21–24, Figs. S50 and 51 and Schemes S13–15.

## Competing reactions

The addition of an AMP handle to the functional nicotinamide group could harbor an advantage for specific reduction in a mineral-based environment. To test this hypothesis, we conducted experiments with both NMN and NAD$^+$ in the same reaction mixture using µNi and µFe as metal promoters at pH 8.5 to explore the reduction of both cofactors in direct competition (Table 1; Supplementary Scheme S15, Fig. S53 and Table S24). As controls, we reduced NAD$^+$ and NMN separately. For the mixed experiments, both cofactors (12 mM ea.) were combined with 600 mM of metal powder, leading to a 25:1 metal to cofactor ratio. In all cases, the 1,4-NADH concentration exceeded that of 1,4-NMNH (Table 1). The results indicate that NAD, while delivering comparable reduction yields for itself in all experiments, seems to have a dampening effect on NMN (over)reduction when both cofactors are in the mixture.

Ultimately, reducing NAD and NMN with the help of H$_2$ and metal catalysts is just one part of these cofactors' role in the prebiotic path towards the first functioning cells—being able to act as a reductant is equally important.

## The reduction capability of NMNH and NADH

Investigating the redox potential of both 1,4-NADH and 1,4-NMNH standards via cyclic voltammetry (starting from an anodic current) helped compare their reduction potential with that of the reaction mixtures of nNiFe-assisted reduction of NAD$^+$ and NMN with H$_2$ (Supplementary Table S18, Figs. S46 and 47). In the case of NiFe-assisted NAD$^+$ reduction, the resulting mixture shows only the reduction potential of 1,4-NADH, while in the case of NMN reduction, the reduction potential of both 1,4-NMNH and that of a second reduced species (most likely the species with the second highest concentration, **2c**) is measured. As the second anodic peak has a more positive reduction potential (meaning is harder to oxidize), the 1,4-NMNH species is the most relevant reductant, not only in a biological but also in a prebiotic context[59–61]. One could argue that it is possible that the 1,4 position of an overreduced species would show a similar reduction potential as a single reduced 1,4-species. However, as the oxidation of the latter leads to the aromatization of the nicotinamide ring, this reaction would be energetically favorable. This theoretically also applies to the single-reduced 1,6-NADH, but we could not isolate this side product to test it as we did for 1,4-NADH and 1,4-NMNH.

It was recently shown that Fe(III) ions (among other metal ions and also minerals) can promote the reaction of 1,4-NADH with pyruvate to lactate abiotically (Supplementary Scheme S16)[11]. Here, we used this reaction as a proxy to compare the reducing capabilities of 1,4-NMNH and 1,4-NADH, showing that both molecules can reduce pyruvate to equal amounts under aqueous conditions with Fe(III) in 17 h at 40 °C (pH < 5; Supplementary Figs. S54 and S55, and Table S25), based on recently published experiments by Mayer and Moran[11]. These results underline that both 1,4-

NMNH and 1,4-NADH are equally good hydride donors and thus that the adenosine nucleotide tail does not – or at least not strongly – influence the efficiency of the catalyzed hydride transfer (Fig. 5).

The conditions used for reduction and for oxidation in this paper diverge, while for reduction, slightly alkaline conditions are used, oxidation is conducted under acidic conditions. 1,4-NADH is known to hydrolyze under acidic conditions[62]. We performed qualitative experiments at pH 5.5 over µFe and µNi with both NAD$^+$ and NMN (Supplementary Scheme S17 and S18, Figs. S56 and S57) to confirm this applies to both nicotinamides in a similar manner. These experiments show the formation of hydrolysis products of the 1,4-species of both cofactors in the case of Ni, while, over Fe, also 1,4-NMNH and 1,4-NADH can be detected, probably due to the increase in pH (up to pH 8) during the latter experiments. Although NAD$^+$ and NMN will be reduced under acidic conditions, they are hydrolyzed quickly. The oxidation of 1,4-NADH and 1,4-NMNH, however, seems to work preferably under acidic conditions[11].

## Discussion

In this study, we have shown that both nicotinamide mono- and dinucleotide can be reduced under conditions found in serpentinizing systems, i.e., with H$_2$ gas promoted by Fe and Ni containing minerals. Relative to NMN, the presence of a second nucleotide (in the form of AMP) in NAD$^+$ influences the reduction product spectrum associated with the nicotinamide ring. We demonstrated that NMN is much more reactive than NAD$^+$ in a time course experiment with NiFe$_3$ nanopowder. Within 1 h, a lot more NMN is consumed than NAD$^+$ in 4 h, under the same experimental conditions. The first and main product of both reactions seems to be 1,4-NADH/NMNH. However, while 1,4-NADH remains stable in solution, 1,4-NMNH quickly undergoes further reduction to form increasingly reduced products. From previous studies[12], we know that 1,4-NADH is not overreduced and remains stable even when the experimental conditions are more reducing or a higher metal to cofactor ratio is employed.

Where does this specificity for 1,4-NADH come from? It is known and well-described that NAD(H) in aqueous solution alternates between a folded (Fig. 6) and open conformation[51–55]. This could shield the nicotinamide ring from excessive overreduction and possibly also from side reactions such as hydrolysis. Here, we performed molecular dynamics calculations confirming that NAD$^+$ not only can bind to a Ni surface in a folded conformation, but also that it lingers less at the surface once reduced to 1,4-NADH than 1,4-NMNH does. These observations support that the AMP moiety plays a role in the specificity of NAD$^+$ reduction, but further investigation will be necessary to understand the very details of the mechanisms in question.

The role of Fe and Ni individually in NMN reduction revealed that Ni tends to generate overreduction products in NMN reduction, while Fe promotes the formation of 1,4-NMNH, the second main side-product being

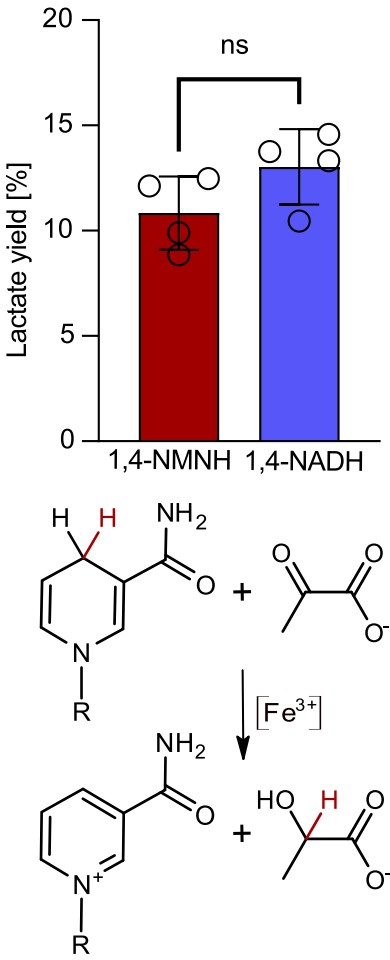

**Fig. 5 | Reduction of pyruvate to lactate with 1,4-NMNH (n = 4) and 1,4-NADH (n = 4).** The method of the reaction was performed according to Mayer and Moran[11] with the help of Fe(III) chloride as a catalyst in 17 h. Both nicotinamides perform equally well in this reaction. In both cases, lactate forms in comparable amounts (for original NMR spectra and quantification Supplementary Figs. S54 and S55, and Table S25). Unpaired $t$-test (two-tailed $P$ value = 0.1303) shows the difference between NADH and NMNH-dependent pyruvate reduction to be not significant (ns). Circles represent individual data points, error bars show SD.

the hydrolysis product **2b**. However, experiments with a lower Ni-to-cofactor ratio (e.g., Table 1, row 1) reveal that Ni also promotes the formation of **2b**, suggesting that the hydrolysis product forms whenever 1,4-NMNH is not overreduced very quickly.

Sebastianelli et al.[9] showed that from a variety of $NAD^+$ metabolites, NMN and $NAD^+$ are the most reactive in seawater. They propose that the phosphate group in NMN repulses negatively charged molecules in solution from reacting with it. In the case of NAD, the negative charge is distributed among the two phosphate groups, so it reacts better. When $Mg^{2+}$ or other relevant cations are present, such as in seawater, they stabilize the negative charge of the phosphate groups of $NAD^+$ and NMN, increasing reduction yields significantly. For heterogeneous catalysis, the relevant interactions seem to happen at the nicotinamide ring. In the presence of metal-bound hydrides[63] the unsaturated hydrocarbons are absorbed, which means hydrogenation can happen over the entirety of the ring (Supplementary Scheme S10)[64–66]. This is further backed up by the early presence of **2d**, the fully hydrogenated form of NMN, in Ni-assisted reactions, as this indicates a direct association with the mineral surface of the nicotinamide ring.

Based on these results, one can discuss how environmental conditions such as metal availability could have influenced the prebiotic selection process of redox cofactors. The stabilization of NAD's functional

nicotinamide part by the AMP moiety means that the redox properties of NAD could have been maintained within a broader variety of environmental conditions than without the moiety. We validated this hypothesis further by performing experiments with both NMN and $NAD^+$ in the same reaction mixture using μNi and μFe as metal promoters. Here, the concentration of 1,4-NADH always surpasses that of 1,4-NMNH, the latter forming more side products. In addition, NAD decreases NMN (over) reduction, lowering the interaction of NMN with the metal surfaces and metal-cofactor interactions (Table 1).

Cyclic voltammetry experiments showed that the oxidation potential of single-reduced species, while all further side (overreduction) products fall behind. Concerning the single-reduced side product 1,6-NADH, we assume it to have a comparable redox potential as 1,4-NADH, although we cannot account for possible steric hindrances during actual reduction reactions. In a biological context, only oxidation at the 1,4-position of the nicotinamide ring is observed. Prebiotically, a 1,6-species could also be relevant for reduction, but has not yet been reported in an experimental setup. The higher reducing strength of single-reduced nicotinamide species creates a mechanistic bottleneck for the back reaction as both overreduction and hydrolysis products could likely not compete as reducing agents in a prebiotic scenario. 1,4-NADH has been shown to perform abiotic reduction of pyruvate under acidic and neutral[11] and reductive amination of pyruvate under alkaline[27] conditions. Here, we have shown here that 1,4-NMNH acts equally well as a hydride source in a non-enzymatic context.

These results substantiate how the AMP moiety can be essential for the targeted reduction of the 1,4-position, as well as for the stability of this specific reduction product. In other words, NAD is functional in a wider variety of environments than NMN, ensuring specific nicotinamide reduction and thus maintaining a steady redox potential in the form of single-reduced NADH. Assuming that redox cofactors present a way to detach hydrides from a mineral surface under certain conditions[29] to expose them to different environmental conditions, a molecular structure stabilizing the optimal reducing form (NAD) would be preferable over one that does not (NMN, Fig. 6).

As NAD effectively detaches hydrides from mineral surfaces, it enables the transport to other geochemical conditions and thus a separation of conditions for reduction and oxidation. Necessary fluctuations, e.g. in pH, would thereby be a natural mechanism to facilitate the role of organic hydride carriers (Fig. 6). The conditions tested here were designed after serpentinizing systems that exist in both acidic and alkaline conditions, though geologically separated from each other[40,41]. Alternating physico-chemical conditions on a micro-compartment level within serpentinizing systems have been observed and further hypothesized as a driving force for prebiotic reactions[67–70]. The real effect of such alternations, regardless of the exact geochemical system, still needs to be investigated in both laboratory and natural settings.

In summary, our findings suggest an evolutionary rationale behind the tenacious conservation of the AMP handle in NAD. Its presence reflects a prebiotic functional constraint that mediated the specific reduction of the hydride carrier under environmental conditions where $H_2$ was the electron donor, made accessible via mineral surfaces. If the first nicotinamide-dependent enzymes arose in such an environment, they would have required the AMP moiety not as a handle, but as an inherent structural property of the NAD cofactor that permitted its function with $H_2$ as the reductant on metal catalysts. In that sense, AMP in NAD is not so much a handle as it is an insulator that protects the cofactor from overreduction. Whether other nucleotides such as guanosine monophosphate (GMP) could have a similar effect when associated with NMN remains an open question. What we do know is that adenosine-derived tails are the common denominator of several cofactors with diverging functions (such as FAD, CoA or SAM). It seems feasible that also for them the extended structure could have been of merit in a prebiotic setting prior to a biological function[18,71].

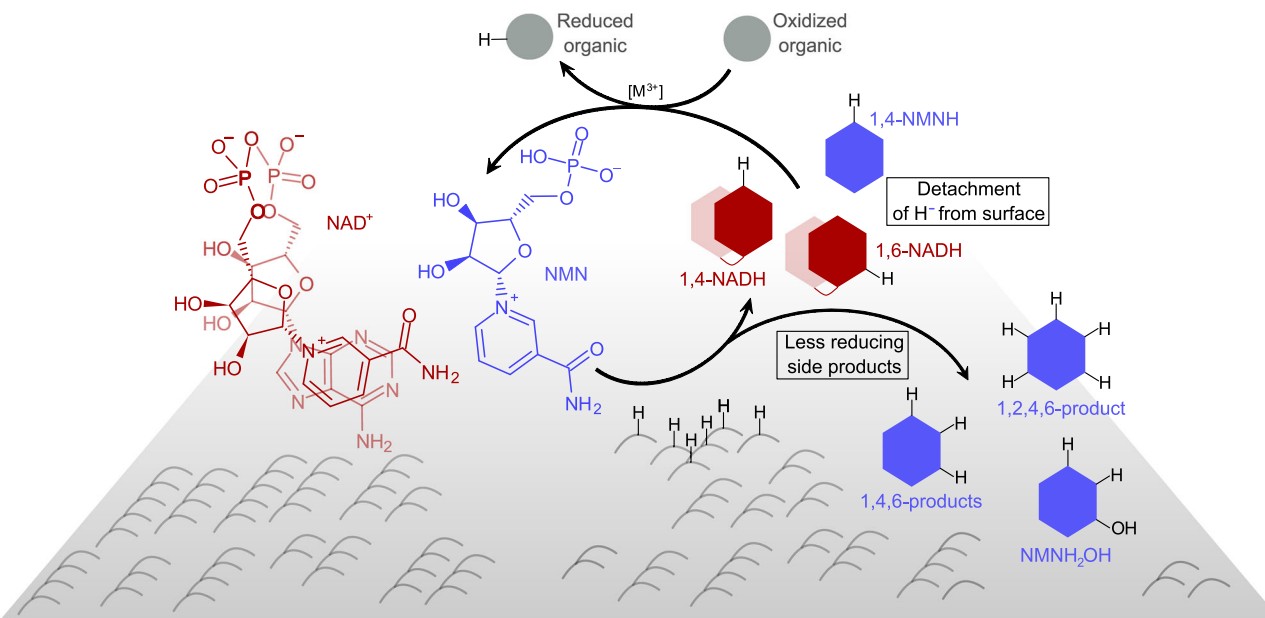

**Fig. 6 | Summarizing figure showing all educts and products detected during simultaneous reduction of $NAD^+$ and NMN.** NMN produces more side products than NAD, possibly creating a mechanistic bottleneck for the back reaction, for which only one-time reduced nicotinamides have a comparable reducing strength. Both overreduction products such as products and hydrolysis products such as $NMNH_2OH$ could likely not compete as reducing agents in a prebiotic scenario. Consequently, only single-reduced species such as 1,4-NADH and 1,4-NMNH could be coupled for reduction reactions, thus detaching the reduction from a mineral surface. Based on assessments of the redox potential, we assume 1,6-NADH to be able to reduce equally well as 1,4-NADH (not accounting for possible steric hindrances).

## Table 2 | Standard metal preparation

| Metals | MW | MW/ metal atom | 18 μmol (mg) | 36 μmol (mg) | 1.8 mmol (mg) |
|---|---|---|---|---|---|
| Fe | 55.85 | 55.85 | 1.01 | 2.01 | 100.5 |
| NiFe$_3$ | 226.24 | 56.56 | 1.02 | 2.04 | |
| NiFe | 114.54 | 57.27 | 1.03 | 2.06 | |
| Ni$_3$Fe | 231.92 | 57.98 | 1.04 | 2.09 | |
| Ni | 58.69 | 58.69 | 1.06 | 2.11 | 105.7 |

The molecular weight (MW) was normalized to the number of metal atoms in each metal powder. The mg of metal used in experiments with 18, 36, and 1800 μmol are listed for each metal used.

## Materials and methods
### Metal preparation
The synthesis of Ni$^0$, Fe$^0$, and Ni-Fe nanoparticles was carried out as described by Beyazay et al.[38]. Commercially bought Fe$^0$ (reduced, <149 μm, Carl Roth, referred to as μFe$^0$) and Ni$^0$ micropowders (3–7 micron, Thermo Scientific, referred to as μNi$^0$) were treated under 5 bar of H$_2$, at 50 °C for 16 h before being used. A detailed characterization of these metals can be found in the same publication.

### Experimental setup with Ni–Fe alloys
Under anaerobic conditions, using a glovebox (JACOMEX), 3 mL of anoxic 0.133 M phosphate buffer solution (PBS; pH 8.5; potassium phosphate monobasic and sodium phosphate dibasic, Sigma-Aldrich, in HPLC-grade water); bubbled with N$_2$ for 1 h containing 36 or 18 μmol of the organic nicotinamide (Nam) compound NAD$^+$ (>95.0%, TCI) or NMN (100% Uthever, MoleQlar; >98.0%, TCI; Supplementary Fig. S58) were placed in 5 mL glass vials (beaded rim) with a polytetrafluoroethylene (PTFE)-coated stirring bar. Equimolar amounts (relative to the cofactor) of metal atoms of Fe, NiFe$_3$, NiFe, Ni$_3$Fe, or Ni nanopowders were added to the bottom of each vial, with the exception of a metal-free control (Table 2). Alternatively,

experiments with Fe and Ni micropowders had 1.8 or 3.6 mmols of metal and 36 or 18 μmol of cofactor (metal-cofactor ratio 50 or 200:1, respectively) in 0.5 M PBS (pH 8.5 or pH 5.5). The vials were sealed with a crimp cap with a PTFE-coated membrane. To allow gas exchange between the interior and the exterior of the glass vial, a syringe needle was inserted through the crimp cap membrane before the vials were placed in the high-pressure reactor. Control experiments with 0.75 M carbonate buffer (sodium bicarbonate, Fisher Scientific; sodium carbonate, Thermo Scientific, 99.5% extra pure) were prepared similarly to other μFe experiments with a 50:1 ratio, described above. To reach the desired pH of 8.5, it was adjusted with concentrated HCl (~37%, Fisher Scientific).

### Standard high-pressure reaction
After pressurizing the reactor (Berghof Reactor 300) with either 5 bar of Ar gas (99.999%, Air Liquid) or 5 bar of H$_2$ gas (99.9% Nippon Gases), the reactions were started and regulated by a controlled reactor heating system (Berghof Products + Instruments). Reactions were performed from 1 h to 4 h at 40 °C and 400 rpm, in a Berghof Reactor Heating System (BR-HS). Afterward, reactors were depressurized under anaerobic conditions and the samples (metal powders and solution) were transferred to 2 mL Eppendorf tubes and centrifuged for 20 min, at 4 °C, and 13,000 rpm (Fresco 17 Microcentrifuge). The supernatants were subjected to different analyses, which are described below.

### Reduction of pyruvate with Fe(III) and 1,4-NMNH or 1,4-NADH
These experiments followed the protocol described in Supplementary Tables S15 and S16 of the paper Mayer et al.[11]. An aqueous mixture of 0.1 mL with 0.1 M pyruvate (Pyruvic acid, Carl Roth), 0.2 M 1,4-NADH (95%, Thermo Scientific), and 0.06 M FeCl$_3$ (98% anhydrous, Grüssing GmbH) reacted overnight at 40 °C and 400 rpm (pH < 5). For the removal of metal ions, it was added 0.2 mL of a thiolate/phosphate solution (100 mg NaSH, 100 mg NaOH in 10 mL saturated aqueous Na$_3$PO$_4$), and left to settle in the fridge (4 °C) for 3 h. Instead of a DMSO standard as used in the referenced protocol, 0.1 mL of a 7 mM DSS stock solution was added at the end of the experiment. To reach a certain volume, 0.2 mL of D$_2$O was also

added before the sample was measured. [1]H-NMR spectra were obtained by an AV III HD 250 MHz Spectrometer with a Double Resonance Broad Band (BBOF) probe head. The same experiment was repeated with 1,4-NMNH (97%, AmBeed) instead of 1,4-NADH.

## Quantitative proton nuclear magnetic resonance (qNMR) analysis

To monitor reactions, as well as detect and quantify the formation of reduced NADH and side products, we established a protocol for quantitative proton nuclear magnetic resonance ([1]H-NMR)[72,73]. The internal standard was a 7 mM solution of sodium 3-(trimethylsilyl)-1-propanesulfonate (DSS, $CH_3$ peak at 0 ppm; >98.0%, TCI) in deuterium oxide ($D_2O$ 99.8 atom %D, AcroSeal, Thermo Scientific), mixed 1:6 with the supernatant of our samples. qNMR spectra were obtained on a Bruker AVANCE-NEO 600 MHz spectrometer equipped with a 5 mm iprobe TBO with z-gradient. Thirty-two scans were made for each sample with a relaxation delay of 40 s (600 MHz) and a spectral width from −3 to 13. Analysis, deconvolution (Supplementary Tables S10–12 and Figs. S21–24) and integration were performed using MestReNova (v.15.0.1). Metal-free controls (ran under the same conditions as the quantified, metal-containing samples) were used as references to the initial amount of NAD/NMN in the sample to account for evaporation and possible degradation under the given pH, temperature and time. The average initial amount of cofactor in the controls was used as $t = 0$ h, and to normalize the reaction yields.

## Standards

[1]H-NMR standards were prepared with 24 mM of the compound and 1 mM of DSS dissolved in $D_2O$ (Supplementary Fig. S59). The spectra were obtained by an AV III HD 250 MHz Spectrometer with a BBOF probe head.

## Product characterization through 2D-NMR

2D [1]H-NMR enabled the assignment of peaks for 1,4-NMNH and NMN in accordance with literature and in comparison to the pair NAD/NADH (Supplementary Tables S26 and S27, Figs. S60–74)[74]. Reduction products were also characterized through different 2D-NMR correlation spectra (Supplementary Scheme S4 and S7, Figs. S10–19 and S26–29). 3 mL of sample from a 1 h and 4 h reduction of NMN with equimolar amounts of $NiFe_3$ (5 bar $H_2$, 40 °C, 400 rpm) were dried using a vacuum concentrator (SpeedVac DNA 130, Savant). The remaining solution and pellet were suspended in 500 µl of $D_2O$ to increase the concentration of the products and resolution of the NMR spectra. The same procedure was performed for a $NAD^+$ sample after a 4 h reaction with equimolar amounts of $NiFe_3$ (5 bar $H_2$, 40 °C, 400 rpm). Two-dimensional correlation spectra of [1]H, [1]H DQF-COSY (Double-Quantum Filtered COrrelated SpectroscopY), [1]H, [1]H TOCSY (Total COrrelated SpectroscopY),[1]H, [13]C HMBC (Heteronuclear Multiple Bond Correlation spectroscopy) were recorded with standard pulse programs[75]. Edited HSQC (Heteronuclear Single Quantum Coherence spectroscopy) spectra were recorded using sensitivity improvement with echo/anti-echo gradient selection and multiplicity editing during the selection step[6,7]. NOESY (Nuclear Overhauser Effect SpectroscopY) spectrum was recorded with mixing time of 1.5 s. Chemical shifts are referenced with the sodium salt of trimethylsilyl-propanesulfonic acid (DSS). Spectra were obtained with the same instrument as qNMR and compared to a list of possible products.

## Data availability

The data that support the findings of this study are available in the SI Appendix. Original analysis files (LC–MS, NMR) will be provided by the corresponding authors upon reasonable request.

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

## Acknowledgements

D.P.H.P. and M.P. thank Bill Martin and Andrea Rosellyn for critical reading and discussions. D.P.H.P. and M.P. thank Alicia Casitas and team for providing access and help to cyclic voltammetry measurements. X.X. thanks Armin Geyer for discussions and the DFG funding for the NMR spectrometer NEO600 (Forschungsgroßgeräte project number 508097909). The authors acknowledge the use of resources provided by the Isambard 3 Tier-2 HPC Facility. Isambard 3 is hosted by the University of Bristol and operated by the GW4 Alliance (https://gw4.ac.uk) and is funded by UK Research and Innovation and the Engineering and Physical Sciences Research Council (EP/X039137/1). M.P. thanks the Max Planck Society (MPG), the International Max Planck Research School 'Principles of microbial life' and the Human Frontiers Science Program (RGEC29/2025) for funding. H.T. thanks MPG, the Volkswagen Foundation (96_742) and Deutsche Forschungsgemeinschaft (TU 315/8-1 / TU 315/8-3). S.V.S. thanks the NERC Doctoral Training Partnership grant (NE/S007407/1) for funding of her Ph.D. Project. This project was supported by the European Regional Development Fund (ERDF) and the Recovery Assistance for Cohesion and the Territories of Europe (REACT-EU).

## Author contributions

Conceptualization: M.P. & D.P.H.P. Methodology: M.P. & D.P.H.P. Investigation: D.P.H.P., M.P., Z.S., S.V.S., T.B. Validation: D.P.H.P., S.V.S., Z.S. & X.X. Formal analysis: X.X., S.V.S., J.B., N.P., D.P.H.P. & M.P. Resources: H.T., V.E., K.V. Writing Original Draft: M.P. & D.P.H.P. Writing Review & Editing: all authors. Visualization: M.P., D.P.H.P. & X.X. Supervision: M.P.

## Funding

## Competing interests

The authors declare no competing interests.
