## [Transparent Peer Review file · Communications Chemistry]

Reduction of NAD and NMN on mineral surfaces with H₂ reveals a functional role for the AMP moiety in prebiotic evolution

Corresponding Author: Dr Martina Preiner

Version 0:

Reviewer comments:

Reviewer #1

(Remarks to the Author)

The manuscript from Henriques Pereira et al. adds to a growing number of studies that have probed the potential role of NAD⁺ in prebiotic chemistry. The manuscript purports that NAD⁺ is unique among nicotinamide-containing molecules in not being over-reduced based on studies with powders of metal and H₂. Analysis was mostly by NMR spectroscopy. The work is thought provoking, and I do not doubt that technical skills of the authors. However, it seems to me that too much is concluded based on the comparison of only two molecules, which in this case was NAD⁺ and NMN. The work also did not adequately discuss similar previous studies.

The following statement in the abstract is made: "Is it a biochemical 'handle' that allows proteins to bind the cofactor more tightly, or is it a relic from a time when cofactors arose from the building blocks of genes?" Here, "it" refers to adenosine. What is surprising is that there is a previously published paper that explicitly discusses and probes whether adenosine is a handle or not. This paper was cited (reference 9, Sebastianelli et al., Chemistry, 30, e202400411, 2024) for the non-enzymatic activity of NAD⁺ and the role of the phosphate groups of NAD⁺, but the adenosine handle part of the work was completely ignored. What is even more surprising is that the necessity of adenosine was not thoroughly investigated in the submitted manuscript from Pereira et al. Certainly more molecules could have been tested than just NAD⁺ and NMN before deciding on the role of the adenosine moiety. To be clear, the experiments in the submitted manuscript are different, but comparison of the two papers would be much more insightful than ignoring the previous study.

Similarly, a little more discussion of reference 27 (Nogal et al. Org. Chem. Front., 11, 1924–1932, 2024) would be warranted, too.

The difference between NAD⁺ and NMN is an AMP unit, as the authors state on page 5. That's a bit more than a nucleoside, which does not have a phosphate. What seems to be tested in this manuscript is the effect of an AMP nucleotide, not an adenosine nucleoside.

I am concerned about the role of free phosphate on the described chemistry. Frequently phosphate is found to participate in prebiotic chemical reactions, and this manuscript used phosphate buffer (0.133 M at pH 8.5 & 0.5 M at pH 5.5). Repeating one of the reactions with another buffer to demonstrate that phosphate was not participating in the reaction would be important, in my opinion. Also, this buffer was used a bit outside of its typical buffering range.

"NAD⁺ reduction at 40 °C over 4 h with equimolar amounts (normalized to 1 metal atom per NAD molecule)..." Could the authors describe a bit better what is meant here? How was the normalization done? I apologize if I've missed something that is already in the manuscript. Incidentally, the data in this figure is nice/interesting. Is the ratio of 1,6-NADH and 1,4-NADH the same for all alloys? The text wasn't very clear to me regarding this.

The fact that NMN is susceptible to over-reduction with the NiFe-alloy, while NAD is not, is a noteworthy result. However, this effect was not very strong when iron alone was used. As the number of iron-containing enzymes is much greater than the number of iron-nickel enzymes in biology, this may suggest that there isn't much of a problem if the "adenosine handle" is missing. Isn't that also supported by what's known regarding prebiotic quantities of iron and nickel? Or do these result

suggest specific environments rich in nickel-containing minerals?

"After quantification of all identified species, we can account for at least 70.7% of transformed NMN for all reactions, often more." The value was 87.4% for NAD⁺. That's impressive for a non-enzymatic reaction.

"In that sense, adenosine in NAD is not so much a handle as it is an insulator that protected the cofactor from overreduction." This is an interesting point that may very well be true. My only problem with this is that this hypothesis wasn't really tested. More molecules would need to be evaluated to come to this conclusion.

Reviewer #2

(Remarks to the Author)

Background:

Many important cofactors employed by biology contain an adenosine monophosphate (AMP) moiety, which presumably evolved as a handle for enzymes, but the authors investigate another possibility that AMP, at least in the case of the redox cofactor NAD, could improve the redox properties of the nicotinamide moiety. In this manuscript, Preiner and coworkers characterised the reduction of NAD⁺ and its AMP-free analogue nicotinamide mononucleotide (NMN) using mineral surfaces consisting of different combinations of nickel and iron with H₂ gas, and they used NMR spectroscopy to identify and quantify reaction products. The authors find that NMN⁺ gets over-reduced mostly to NMNH₃ and NMNH₂OH, while NAD⁺ mostly gets reduced to NADH. The authors propose that the AMP moiety prevents NMN from over-reduction by forming a folded structure where the adenine ring is stacked with the reduced nicotinamide ring. This folded structure is not capable of adhering productively to the metal surfaces, inhibiting further reduction. They conclude that the AMP moiety conjugated to NAD may have been selected based on its ability to prevent overreduction, providing an alternative proposal to the handle hypothesis.

Overall, this is an interesting manuscript with a creative premise and reads well. The experiments are well-conducted, but some of the NMR assignments may be incorrect. Publication of the manuscript is recommended after the following comments/criticism have been addressed.

Comment to authors:

1. One of the most interesting ideas in the manuscript is that the folded conformation of NADH is what prevents its over-reduction. The fact that no over-reduced products are detected for NAD⁺ would suggest that the folded conformation of NADH is very stable, with very little of the open conformation existing at equilibrium. However, according to reference 57, at 295 K, the fraction of the folded form is only 50% and this fraction should be even less at 40 C, which was the temperature of the reduction reactions. Hence, most of the NADH should be in the open form under the author's reaction conditions, at least in the bulk solution. It may still be the case that NADH only binds to the metal surface in the folded conformation, but there is no evidence for this presented. The reviewer suggests the authors conduct experiments to test this hypothesis. For example, it may be possible to use solid-state IR or Raman spectroscopy to directly characterize NADH conformation when adhered to the metal surfaces. Another possible experiment is to add a sterically bulky group to the adenine nucleobase that should prevent efficient folding and lead to over reduction. Alternatively, the authors may consider conducting computational studies on the interaction of NADH on mineral surfaces to provide support for their proposed mechanism. Obviously, the more studies, the better, but the authors should do at least one before publishing.

2. The NMR the peak assignments to 1,4,6-NMNH₃ in Figure 3A need better justification. Two peaks are assigned to a single proton at the 2 position. It's not obvious why the authors have made such an assignment, and the reviewer could not find any discussion about it in the main text or SI. The assumption would be that this pair of peaks is really a doublet, but there is no reason to expect that the proton at C2 should have such a large coupling constant, given there are no adjacent protons at the C1 or C3 positions. For example, the C2 protons for NMNH₂OH and 1,4-NMNH don't show such a large coupling constant, as would be expected. Even assuming that the proton at C2 of 1,4,6-NMNH₃ really is a doublet as implied in the Figure, then why would the ratio of intensities of the two signals comprising the doublet change over time as shown in the stacked spectra? It's difficult to gauge by eye, but the combined intensities of these two signals don't really seem to change much after 1 h, while the graph in Figure 4D shows the production of 1,4,6,-NMNH₃ increases linearly over time. The authors need to rethink this proton assignment, or provide some sort of justification for it, especially since they are using these signals to quantify yields. A misassignment could lead to very different conclusions about the yields of this compound overtime.

3. Based on Table S10, 1,4,6-NMNH₃ and NMNH₂OH increase from 3 to 4 hours, however, the Figure 3A and S24 indicate the opposite. Is this due to peak broadening or the last spectrum not scaled properly?

4. The authors listed NMR peak assignments and multiplicities for all the standard compounds, but it would be helpful if they also did it for all the reduced NAD⁺ and NMN products as well.

5. For Figure 4B, is there a reason for the inclusion of nano-NiFe instead of macro-NiFe powder? As the authors pointed out, the use of macro-NiFe powder would make the comparison more valid.

Minor comments

1. For Figure 4A (Page 8), in order of decreasing Ni, the pie chart for NiFe₃ should be in the middle.

2. On page 5, replace "In additional to" with "Additionally" or "In addition to".
3. For Figure 3A, the way the NMR spectra are stacked leads to a lot of overlap, which makes it difficult to interpret. The reviewer suggests that the NMR spectra be stacked in a way that avoids significant lap. Also, some of the text in Figure 3B is quite small and difficult to read.
4. In Table S13 (page 40) of the SI, the "0" in μNiO should be superscripted.

Reviewer #3

(Remarks to the Author)

This paper deals with an important question in our understanding of co-factor evolution: why do so many cofactors contain an adenosine moiety? It's a bit of a perennial question which has not been definitively addressed. The current paper provides a fresh and new view: by comparing the reduction of nicotinamide adenine dinucleotide (NAD) (the molecule used as a cofactor in many enzymes) to a truncated nicotinamide molecule which lacks the adinonine, the authors arrive at a novel hypothesis where the latter might be imagined as being involved in mineral binding and the reduction process itself. I find this to be an intriguing proposal, and the data provide can be used to make such an argument, though remaining somewhat speculative.

Below I provide major and minor comments interspersed.

"investigate the ability of naturally occurring Ni- and Fe-containing allows to replace" → "alloys"

"nano-casting method by using tea leaves". Is there a more prebiotically relevant way to make such nanoparticles? Relying on plants for prebiotic chemistry is a little distant.

Figure 2: Please specify what ND is. Is it Not detected, or no data for example?

Page 4: "or by direct electron transfer to NAD" Is there any evidence of a metal directly reducing a nicotinamide? If so what is the intermediate prior to addition of a proton? If there's no evidence, it's best to remove this.

At the top of page 4 there is an interesting discussion comparing data presented in figure 2 but the discussion is quite vague and not definitive. Can the authors do more here? Or if not perhaps simply highlighting what is known and not known. Arrangement of the metal atoms, and not simply the number is likely important as well.

page 5: "Without metals, NMN does not react and remained stable (Supplementary Figures S8 and S9). Under Ar, NMN still got reduced due to the abundant iron in the mineral compound, but more slowly than under H₂ (Supplementary Tables S6 and S7)."

This is a confusing 2 sentences. Please revise for clarity.

Around page 5 I began to ask why is the fist section of the paper with figure 2 present in the paper at all? This is a bit of a repeat of previous work - although valuable would it be better to place this in the supplemental and go straight to comparison of different molecule's reduction tendencies?

Page 5 "The remaining 17% can in part be attributed to nicotinamide formation" What does this mean?

page 5: "and a variety of products was observed in" → "were"

page 5: "the concentration of twice reduced 2c increased with once reduced 1,4-NMNH decreasing" please revise this was not clear.

page 5: "Under Ar, 2d did not form at all, " was this or without metal? and with which one if there was a metal? This is a recurring confusion in the paper. Can the authors consider a naming scheme to deal with this. For example Ar(NiFe) or something such that the presence of a metal is more clear?

page 6: "total amount of reduced NMN remained relatively stable". It might be helpful to add a line showing this in the plot.

Figure 3B. Is there any evidence for the 6e- reduction of of NMN without an intermediate? if not, it is best to remove this. The current data seems to lack the required time resolution to show this...

Figure figure legend: ' Full arrows represent proposed reaction mechanisms" Perhaps mechanism implies an understanding of how the electrons and atoms are moving - maybe delete it and simply call them 'proposed reactions' ?

page 7: "As both nNi and nFe visibly" please introduce the "n" abbreviation as nomenclature earlier in the paper, perhaps near the tea leaves casting method. Are the data associated with this section comparable to figure 2 in having used the "n" form? Please label consistently.

Figure 4: what was the amount of time? Are there time course data?

page 8: "The metal-cofactor ratio was 200:1 to guarantee the detection" This change makes it difficult to compare to anything which was above (a 1:1 ratio). I am always loath to propose new experiments but this is an important part of the paper and the variation in the ratios used in the experiments makes the reading and conclusions fragmented. If possible please investigate the ratio dependence.

Page 8: "This could also explain the fast formation of its fully reduced product 2d shown in Figure 4C. NAD in a staggered formation could only absorb partly on the surface, avoiding overreduction" But the NiFe₃ seems to make 1-4NADH quite well?

Paragraph at bottom of page 8 and going into page 9: This is interesting but not discussed in a substantial way: why is this? If the authors don't want to elaborate it could be cut.

page 9: "seems to have a dampening effect on NMN (over)reduction when both cofactors are" This was an important experiment and provides an interesting result. It's related to the above brief discussion of why Ni and Fe seem to reduce differently. It's notable that the amount of 1,4NADH is lowered in the mixed experiment compared to the unmixed (13.6% vs 19.8%). Perhaps there is some contribution from the metal and the organic..

in table 1: the individual co-factor concentrations were both 12mM if I understand it, perhaps that can be indicated more clearly in the table which currently simply says 12mM...

Also in the table: is a two-tailed comparison the correct one here? I'd check back to multiple comparisons - Bonferroni corrections / anova here.

page 9: "As the second signal has a lower oxidation potential" we almost always write "reduction potential" because the directionality of the reaction (what is substrate, and what is product) enters into the Nernst equation. Please revise the paper to discuss reduction potentials, and when referring to the c.v.'s the use of "cathodic" and "anodic" currents can be used to indicate current detected at negative becoming or positive becoming potentials. This made the text difficult to interpret, for example "Cyclic voltammetry experiments showed that the oxidation potential of single-reduced species, while all further side (overreduction) products fall behind. Concerning the single-reduced side product 1,6-NADH, we assume it to have a comparable redox potential as 1,4-NADH, although"

Please consider a way to write this using reduction potential, cathodic, and anodic..

The performance of the electrochemistry experiments is really great, but the potentials reported seem completely off with our knowledge:

For example the Cyclic voltammetry starting at S30.

The reduction potential of NAD⁺ is ~-580mV vs SCE, but the CV shows something very different with the cathodic current appearing at about -1V vs SCE. This is very negative: what is it? It is negative enough that it could be H⁺ reduction at the electrode.

in S30B, the anodic current appears at +500mv vs SCE. This is very positive and more coincident with reactions involving oxides/O₂. In table S12 the listed "oxidation potentials" (which I presume are the potentials associated with the major cathodic currents in the c.v.) are also in this regime, which is outside of the redox space of nicotinamides.

(back to the main text)

page 12: "The immediate presence of 2d" as with an above comment, the time course data does not seem at a small enough increment to show this.

in S14, can the authors provide the expected position of the methine carbon they are referring too and a picture of that dimer molecule to aid in interpretation? This figure lacks reference to in the SI as well.

Final comment: The authors used phosphate buffer in their experiments. Metals and phosphates interact strongly, and the interaction between Fe and Ni is different. Thus one could make hypotheses that the differences observed in the reductive tendencies is due to a blanket of phosphate on the minerals. I strongly encourage the authors to perform experiments where this can be investigated prior to publication. One might consider carbonate buffer due to its relevance in natural systems, comparing reduction products from NAD vs NMN with the metal alloys used and also pure Ni and Fe. Without some isolation of the effect of phosphate we are left wondering about its importance.

Version 1:

Reviewer comments:

Reviewer #2

(Remarks to the Author)

The reviewer thanks the authors for taking the time and effort to respond to the comments regarding the NMR assignments and folding dynamics. The authors have now provided a suite of new NMR and computational data to back up their arguments.

With respect to the new computational data, the reviewer has no further questions or comments. This study is very well done and backs up their hypothesis.

When it comes to the NMR data, the reviewer still has some comments/suggestions. With respect to the two NMR resonances assigned to the C2 proton of 1,4,6-NMNH₃, the authors now suggest these belong to two stereoisomers (more specifically, conformers) brought about by rotation of the amide. This assignment could very well be correct, but the reviewer is still skeptical. Restricted rotation about this C-C bond to the amide might indeed be brought on by conjugation with the double bond, but why then would similar conformational isomerism not be observed with 1,4-NMNH and 1,4-NMNH₂OH? The authors have already carried out quite a large number of NMR experiments, and the reviewer isn't asking for more. The only way to really be sure about these assignments is to make a pure standard of 1,4,6-NMNH₃ (as well as the other compounds), but this may be outside the scope of the present study. The reviewer suggests that the authors only propose that two stable conformational isomers of 1,4,6-NMNH₃ exists, and not be so explicit about what each one is. For example, isomers involving two conformations of the piperidine ring seem possible as well, similar to those seen for substituted cyclohexene rings (e.g., see <https://pubs.acs.org/doi/pdf/10.1021/cr60308a004>). Is there similar literature the authors can cite for piperidine systems? The reviewer is not trying to be overly pedantic, but the NMR spectra are quite complicated, while the proposed products may also be conformationally dynamic giving rise to further spectral complexity. It would be incredibly easy to make a misassignment, and the reviewer, for the authors' own sakes, wants to make sure all the possibilities have been thoroughly considered.

In the case of NMNH₂OH, the authors argue that hydration will result in two diastereomers, cis and trans with respect to the ribose ring. This makes sense, but why then does one diastereomer seem to decrease as the reaction goes on longer? Since the authors have 1,4-NMNH available as a standard, what happens when they dissolve it in water under the experimental conditions except without hydrogen gas? Does it readily undergo hydration? Are two diastereomers observed? This is a simple enough experiment to do that will support these NMR assignments.

Some additional considerations when it comes to stereochemistry are with compound 2d. Reduction should lead two diastereomers of 2d as well, with the amide being either cis or trans with respect to the ribose ring. Is there any evidence of two diastereomers in the NMR spectra?

At the end of the day, since the main point of the manuscript is that NAD⁺ is only reduced to NADH, while NMN is completely reduced, the NMR data definitely support this main conclusion. The reviewer thinks publication is still warranted as long as the authors include a little more discussion of the NMR data for the NMN experiments in the main text. Specifically, they should have at least one paragraph dedicated to how assignments were made, and acknowledge that given the complexity of the spectra, some assignments are tentative.

Reviewer #3

(Remarks to the Author)

The authors have improved the manuscript in the review process. One final suggestion is to include the Meyer paper <https://doi.org/10.1038/s41598-023-49021-4> as a citation and use it to mention the reduction potentials; this would be helpful for readers.

Reviewer #1 (Remarks to the Author):

The manuscript from Henriques Pereira et al. adds to a growing number of studies that have probed the potential role of NAD⁺ in prebiotic chemistry. The manuscript purports that NAD⁺ is unique among nicotinamide-containing molecules in not being over-reduced based on studies with powders of metal and H₂. Analysis was mostly by NMR spectroscopy. The work is thought provoking, and I do not doubt that technical skills of the authors. However, it seems to me that too much is concluded based on the comparison of only two molecules, which in this case was NAD⁺ and NMN. The work also did not adequately discuss similar previous studies.

The following statement in the abstract is made: "Is it a biochemical 'handle' that allows proteins to bind the cofactor more tightly, or is it a relic from a time when cofactors arose from the building blocks of genes?" Here, "it" refers to adenosine. What is surprising is that there is a previously published paper that explicitly discusses and probes whether adenosine is a handle or not. This paper was cited (reference 9, Sebastianelli et al., *Chemistry*, 30, e202400411, 2024) for the non-enzymatic activity of NAD⁺ and the role of the phosphate groups of NAD⁺, but the adenosine handle part of the work was completely ignored.

Answer: Thank you for the kind feedback. We agree that ref. 9 warrants more discussion. However, because Sebastianelli et al. do not show a detailed characterization of their products, it is hard to argue if there is or not overreduction of the molecules tested, thus there are limitations to the comparisons that can be made. A paragraph was added to the discussion section that we hope resolves the criticism and fairly represents both authors' ideas and results:

"Sebastianelli et al. showed that from a variety of NAD⁺ metabolites, NMN and NAD⁺ are the most reactive in sea water. They propose that the phosphate group in NMN repulses negatively charged molecules in solution from reacting with it. In the case of NAD, the negative charge is distributed among the two phosphate groups, so it reacts better. When Mg²⁺ or other relevant cations are present, such as in sea water, they stabilize the negative charge of the phosphate groups of NAD⁺ and NMN, increasing reduction yields significantly. For heterogeneous catalysis the relevant interactions seem to happen at the nicotinamide ring. In the presence of metal bound hydrides the unsaturated hydrocarbons are absorbed, which means hydrogenation can happen over the entirety of the ring (*Supplementary Scheme S7*)."

What is even more surprising is that the necessity of adenosine was not thoroughly investigated in the submitted manuscript from Pereira et al. Certainly more molecules could have been tested than just NAD⁺ and NMN before deciding on the role of the adenosine moiety. To be clear, the experiments in the submitted manuscript are different, but comparison of the two papers would be much more insightful than ignoring the previous study.

Answer: To further test the AMP moiety, we believe it would require molecules that are not easily available, such as Nicotinamide Guanidine Dinucleotide, and NMN with an additional phosphate group. We hope to address this in future work, and that the added discussion and references mentioned above can promote a healthy discussion of the hypothesis as is.

Similarly, a little more discussion of reference 27 (Nogal et al. *Org. Chem. Front.*, 11, 1924–1932, 2024) would be warranted, too.

Answer: We added Nogal et al. to the section of the discussion where 1,4-NADH oxidation is discussed:

“1,4-NADH has been shown to perform abiotic reduction and reductive amination of pyruvate under acidic, neutral and alkaline (Nogal et al.) conditions, respectively. Here, we have shown here that 1,4-NMNH acts equally well as a hydride source in a non-enzymatic context.”

The difference between NAD⁺ and NMN is an AMP unit, as the authors state on page 5. That's a bit more than a nucleoside, which does not have a phosphate. What seems to be tested in this manuscript is the effect of an AMP nucleotide, not an adenosine nucleoside.

Answer: The term was chosen to include all adenosine containing cofactors, which in the case of SAM does not have the phosphate moiety. However, we agree with the criticism that the moiety being tested is in fact AMP, so a few corrections were made, to the title and introduction, to better reflect that (mostly replacing “adenosine” and “adenosine monophosphate” with “AMP”).

I am concerned about the role of free phosphate on the described chemistry. Frequently phosphate is found to participate in prebiotic chemical reactions, and this manuscript used phosphate buffer (0.133 M at pH 8.5 & 0.5 M at pH 5.5). Repeating one of the reactions with another buffer to demonstrate that phosphate was not participating in the reaction would be important, in my opinion. Also, this buffer was used a bit outside of its typical buffering range.

Answer: Good point! But although the phosphate buffer was used outside of its normal range, the pH was monitored to assure that it was still effective. In reactions with very little metal (1:1 ratio to cofactor), the pH was always stable at 8.5, increasing to about 9 when increasing the ratio to 50:1 (metal to cofactor).

Repeating the reactions with μFe in carbonate buffer at pH 8.5 showed that phosphate does not promote any further reactions. In fact, the results obtained with a carbonate buffer showed an almost identical yield of the main reaction products.

Supplementary Tables S15 and S16 are included in this text below (Tables 1 and 2 respectively). The outcome of these experiments were included in the results sections of the maintext:

“To investigate this further, we also performed experiments with Fe and NMN under H₂ in carbonate buffer (Supplementary Scheme S8 and S9), leading to the same yields of as phosphate buffer (Supplementary Tables S15 and S16, and Figures S29 and S30). This indicates that oxidation of Fe and precipitation with the buffer's anions can be a reason for the different outcome of Fe and Ni reactions, but at the same time excludes a catalytic effect of phosphate on the reaction.”

Also, following observation was included into the SI:

“The reduction of NAD⁺ with Fe⁰ in carbonate buffer led to a wider variety of uncharacterizable side products (**Error! Reference source not found.**) than during reactions in phosphate buffer (**Error! Reference source not found.**). The peaks of the side products are less defined than the peak of 1,6-NADH which appears in the same region of the spectra. This effect was also visible in controls without Fe⁰, although in even lesser amounts. The amount of each of the previously characterized main products (1,4-NADH, 1,4-NMNH and NMNHOH), however, is affected.”... “Extra peaks were also found in the NMN carbonate samples, as they were in the samples with NAD⁺, leading to a significant loss of the starting material (**Error! Reference source not found.**; Table 2 bellow), but without affecting the main products yield. In conclusion, phosphate is seemingly less reactive and thus a more reliable buffer.”

Table 1 After 4 h under 5 bar of H₂, as shown in Supplementary Scheme S10 and Scheme S8, samples with μFe⁰ yielded similar amounts of 1,4-NADH, 1,6-NADH, from the starting material NAD⁺, regardless of the buffer used. The starting metal and cofactor were 36 μmol mixed in 3 mL of 0.75 M Carbonate buffer or 0.5 M PBS (pH 8.5). The amount of metal atoms was 50 times the moles of cofactor. The yields were calculated relative to the metal-free sample (100% NAD⁺). To determine the TOF of each reaction, the sum of 1,4-NADH, and 1,6-NADH was considered as the amount of product. PBS samples had duplicates while Carbonate had quadruplets.

	H ₂	NAD ⁺	SD	1,4-NADH	SD	1,6-NADH	SD	Nam	SD	TOF [mol/s]
4h	Carbonate	42,16%	7,1%	7,46%	0,1%	1,90%	0,1%	32,94%	2,8%	1,37E-07
	PBS	57,52%	2,9%	8,12%	1,2%	1,60%	0,2%	27,57%	2,2%	1,43E-07

Table 2 After 4 h under 5 bar of Ar, as shown in Supplementary Scheme 11 and Scheme 9, samples with μFe⁰ yielded similar amounts of 1,4-NMNH, NMNH₂OH, and nicotinamide (Nam), from the starting material NMN. The starting metal and cofactor were 1.8 mmol and 36 μmol, respectively, mixed in 3 mL of 0.75 M Carbonate buffer or 0.5 M PBS (pH 8.5). The amount of metal atoms was fifty times of the cofactor. The yields were calculated relative to the metal-free sample (100% NMN). To determine the TOF of each reaction, the sum of 1,4-NMNH, 1,4,6-NMNH₃, and 1,2,4,6-NMNH₅ was considered as the amount of product. PBS samples were duplicates while carbonate were quadruplets.

	H ₂	NMN	SD	1,4-NMNH	SD	1,4,6-NMNH ₃	SD	NMNH ₂ OH	SD	1,2,4,6-NMNH ₅	SD	Nam	SD	TOF [mol/s]
4h	Carb.	4,52%	3,2%	45,57%	4,5%	0,00%	0,2%	11,58%	1,3%	0,00%	0,0%	10,50%	0,7%	3,44E-07
	PBS	23,48%	15,8%	47,95%	11,9%	0,00%	0,0%	11,43%	2,0%	0,00%	0,0%	14,88%	12,8%	1,72E-07

"NAD⁺ reduction at 40 °C over 4 h with equimolar amounts (normalized to 1 metal atom per NAD molecule)..." Could the authors describe a bit better what is meant here? How was the normalization done? I apologize if I've missed something that is already in the manuscript. Incidentally, the data in this figure is nice/interesting. Is the ratio of 1,6-NADH and 1,4-NADH the same for all alloys? The text wasn't very clear to me regarding this.

Answer: We have moved the supplementary table that clarifies this from the SI to the method section of the maintext. We now also reference it in the maintext where the referee requested clarification. We hope this helps.

The fact that NMN is susceptible to over-reduction with the NiFe-alloy, while NAD is not, is a noteworthy result. However, this effect was not very strong when iron alone was used. As the number of iron-containing enzymes is much greater than the number of iron-nickel enzymes in biology, this may suggest that there isn't much of a problem if the "adenosine handle" is missing. Isn't that also supported by what's known regarding prebiotic quantities of iron and nickel? Or do these results suggest specific environments rich in nickel-containing minerals?

Answer: Thank you for the interesting thought! Notably, when working with NAD (Fig. 2), a combination of both metals seems to be much more efficient than either one alone, we also discuss possible mechanistic reasons for that. To our understanding, the NiFe hydrogenases are the most widespread enzymes in bacteria and archaea and arguably the oldest (*Greening C, Biswas A, Carere CR, Jackson CJ, Taylor MC, Stott MB, et al. Genomic and metagenomic surveys of hydrogenase distribution indicate H₂ is a widely utilised energy source for microbial growth and survival. ISME J. 2016 Mar 1;10(3):761–77*).

Regardless, here, we would not want to extrapolate our results and conclusions up to the enzymatic level, as we are working within the context of non-enzymatic reactions. In the main text, we limit the discussion to prebiotic conditions, considering different metals and minerals, especially the abundant alloy NiFe. Depending on the environment, NMN or NAD could be the more efficient and stable cofactor, with NAD seeming appropriate for a wider variety of environments. As suggested, in environments depleted of Ni, 1,4-NMNH would not be as susceptible to overreduction, but it would still be easily degraded by H₂O to form NMNH₂OH, which NAD is not under the same conditions.

"After quantification of all identified species, we can account for at least 70.7% of transformed NMN for all reactions, often more." The value was 87.4% for NAD⁺. That's impressive for a non-enzymatic reaction.

Answer: Thank you!

"In that sense, adenosine in NAD is not so much a handle as it is an insulator that protected the cofactor from overreduction." This is an interesting point that may very well be true. My only problem with this is that this hypothesis wasn't really tested. More molecules would need to be evaluated to come to this conclusion.

Answer: We think this comment goes into a similar direction as the comment by the same reviewer above. In order to test the exact influence of the AMP moiety, we would need molecules that are not easily available, such as Nicotinamide Guanidine Dinucleotide, and NMN with an additional phosphate group. We have added discussions looking at existing work (Sebastianelli et al. 2024) suggested by the reviewer to evaluate possible interactions further. During revision we have found a way to simulate the behavior of both NMN and NAD on a Ni surface that suggests that folding of NAD is prevalent also on the metal surface. So it seems like a valuable hypothesis that the folding process is indeed an important factor – in the conclusion we also keep room for the idea that any other dinucleotide (NGD) could have a similar effect, we are not arguing our results being a reason for AMP over any other moiety, because we cannot:

“Whether other nucleotides such as GMP could have a similar effect when associated to NMN remains an open question. What we do know is that adenosine-derived tails are the common denominator of several cofactors with diverging functions (such as FAD, CoA or SAM). It seems feasible that also for them the extended structure could have been of merit in a prebiotic setting prior to a biological function.”

We want to address the exact mechanism in future work, but this seems far beyond the scope of this paper.

We hope this resonates with the reviewer and thank the reviewer for a lot of valuable and important feedback!

Reviewer #2 (Remarks to the Author):

Background:

Many important cofactors employed by biology contain an adenosine monophosphate (AMP) moiety, which presumably evolved as a handle for enzymes, but the authors investigate another possibility that AMP, at least in the case of the redox cofactor NAD, could improve the redox properties of the nicotinamide moiety. In this manuscript, Preiner and coworkers characterised the reduction of NAD⁺ and its AMP-free analogue nicotinamide mononucleotide (NMN) using mineral surfaces consisting of different combinations of nickel and iron with H₂ gas, and they used NMR spectroscopy to identify and quantify reaction products. The authors find that NMN⁺ gets over-reduced mostly to NMNH₃ and NMNH₂OH, while NAD⁺ mostly gets reduced to NADH. The authors propose that the AMP moiety prevents NMN from over-reduction by forming a folded structure where the adenine ring is stacked with the reduced nicotinamide ring. This folded structure is not capable of adhering productively to the metal surfaces, inhibiting further reduction. They conclude that the AMP moiety conjugated to NAD may have been selected based on its ability to prevent overreduction, providing an alternative proposal to the handle hypothesis.

Overall, this is an interesting manuscript with a creative premise and reads well. The experiments are well-conducted, but some of the NMR assignments may be incorrect. Publication of the manuscript is recommended after the following comments/criticism have been addressed.

Answer: We thank the reviewer for this kind assessment of our manuscript.

Comment to authors:

1. One of the most interesting ideas in the manuscript is that the folded conformation of NADH is what prevents its over-reduction. The fact that no over-reduced products are detected for NAD⁺ would suggest that the folded conformation of NADH is very stable, with very little of the open conformation existing at equilibrium. However, according to reference 57, at 295 K, the fraction of the folded form is only 50% and this fraction should be even less at 40 C, which was the temperature of the reduction reactions. Hence, most of the NADH should be in the open form under the author's reaction conditions, at least in the bulk solution. It may still be the case that NADH only binds to the metal surface in the folded conformation, but there is no evidence for this presented. The reviewer suggests the authors conduct experiments to test this hypothesis. For example, it may be possible to use solid-state IR or Raman spectroscopy to directly characterize NADH conformation when adhered to the metal surfaces. Another possible experiment is to add a sterically bulky group to the adenine nucleobase that should prevent efficient folding and lead to over reduction. Alternatively, the authors may consider conducting computational studies on the interaction of NADH on mineral surfaces to provide support for their proposed mechanism. Obviously, the more studies, the better, but the authors should do at least one before publishing.

Answer: We really appreciate the interest and feedback of the reviewer!

During the preparation of the manuscript, we tried to observe the different conformations with NMR. To observe it, we would need to cool down the sample, but with water the cooling was

not sufficient. With a different solvent NAD not only would be less soluble, but the molecular dynamics change. Thus, we were unsuccessful in detecting any changes that would indicate the existence of either conformation.

For revision, we established a new collaboration, resulting in a simulation of NAD and NMN under the conditions of our wet lab experiments (Supplementary Fig. S15–S25; part of these figures are below as Figure 1–3). Results show that the folded conformation not only exists at 40°C, but also that both conformations (open and folded) exist as fast (ns) interchangeable states in each molecule, rather than separate forms in solution (Figure 1). The molecules on the Nickel surface can still change conformation, however a lot slower than in solution, a folded molecule stays folded considerably longer than in solution (Figures 2 and 3). We hope this additional data suffices as proof that NAD considerably exists in a folded conformation under our experimental conditions, and support the presented hypothesis that this conformation could influence the reduction pattern.

Figure 1 Folding dynamics of NAD⁺ in water showing: the minimum distance between nicotinamide and adenosine rings (left) and the corresponding number of contacts (right).

Figure 2 Folding dynamics of NAD⁺ with Ni sphere showing a) the minimum distance between nicotinamide and adenosine rings and b) the corresponding number of contacts.

Figure 3 Folding dynamics of NADH with Ni sphere showing a) the minimum distance between nicotinamide and adenosine rings and b) the corresponding number of contacts.

These results are now also incorporated in the according section in the main section, e.g.:

“In order to evaluate these differences further, we performed several molecular dynamics calculations (Supplementary Methods, Supplementary Tables S11 and S12 and Figures S15–25). NAD is known to dynamically fold in aqueous solution. Here, we observe that although NAD interchanges between folded and unfolded conformation in solution, the alternation mostly stops on the metal surface (the simulations were performed on a Ni surface). On the surface, NAD⁺ occurs in stabilized open and closed configurations (Supplementary Figures S16b and S17), with 30–40% being folded. Once reduced to 1,4-NADH, the adsorption to the surfaces decreases (Supplementary Figure 24, Supplementary Table S12).

2. The NMR the peak assignments to 1,4,6-NMNH₃ in Figure 3A need better justification. Two peaks are assigned to a single proton at the 2 position. It's not obvious why the authors have made such an assignment, and the reviewer could not find any discussion about it in the main text or SI. The assumption would be that this pair of peaks is really a doublet, but there is no reason to expect that the proton at C2 should have such a large coupling constant, given there are no adjacent protons at the C1 or C3 positions. For example, the C2 protons for NMNH₂OH and 1,4-NMNH don't show such a large coupling constant, as would be expected. Even assuming that the proton at C2 of 1,4,6-NMNH₃ really is a doublet as implied in the Figure, then why would the ratio of intensities of the two signals comprising the doublet change over time as shown in the stacked spectra? It's difficult to gauge by eye, but the combined intensities of these two signals don't really seem to change much after 1 h, while the graph in Figure 4D shows the production of 1,4,6,-NMNH₃ increases linearly over time. The authors need to rethink this proton assignment, or provide some sort of justification for it, especially since they are using these signals to quantify yields. A misassignment could lead to very different conclusions about the yields of this compound overtime.

Answer: Thank you for this nice comment. As response, assignments are now made clearer in Supplementary Data Figs. SD7 and SD8. The description to this assignment was improved accordingly, as follows:

“Similarly, crosspeaks at 7.448/7.405 – 143.7 (not shown **Error! Reference source not found.**), 3.165 – 39.7, 2.215 – 20.3, and 1.808 – 20.3 ppm were detected in the edited HSQC spectrum (**Error! Reference source not found.**). Relatively resolved ^1H signals at 3.165 ppm (dd, 6.5, 5.0 Hz), 2.215 ppm (t, 6.7 Hz) and 1.808 ppm (m) were observed and could be assigned to three methylene groups, respectively. Connectivities among peaks at 7.448, 7.405 ppm with peaks at 3.165, 2.215 and 1.808 ppm were observed in TOCSY spectrum (Fig. SD17). The tetrahydro-product 2c was identified to be 1,4,6-NMNH₃. The observation of two peaks at 7.448 ppm (s) and 7.405 ppm (s) for H-2 is assumed to be caused by the coexistence of the two stereoisomers as shown in Scheme 1.”

Supplementary Data Scheme SD2 (represented below as Scheme 1) shows the two stereoisomers that best fit the characteristic peaks found for 1,4,6-NMNH₃.

Scheme 1 Stereoisomers of 1,4,6-NMNH₃

Also in the case of NMNH₂OH the occurrence of two peaks for H₂ is assumed to be caused by two stereoisomers. In this case, because the hydration makes carbon 6 a chiral center (Supplementary Data SD3, as Scheme 2 below).

Scheme 2 Stereoisomers of product NMNH₂OH

3. Based on Table S10, 1,4,6-NMNH₃ and NMNH₂OH increase from 3 to 4 hours, however, the Figure 3A and S24 indicate the opposite. Is this due to peak broadening or the last spectrum not scaled properly?

Answer: Our apologies, we messed up the scaling, this is adjusted now.

4. The authors listed NMR peak assignments and multiplicities for all the standard compounds, but it would be helpful if they also did it for all the reduced NAD⁺ and NMN products as well.

Answer: Thank you for the suggestion. Because we could not separate the different products via LC – due to overlapping retention times – we are not able to assign, isolate, and characterize every peak of every molecule. However, we did include assignments with multiplicities and coupling constants to those well resolved peaks, including NMNH, but also peaks of the pyridinium ring of each molecule. We have edited Supplementary Data Fig. SD5, SD7, SD8 and SD10 to include these details.

5. For Figure 4B, is there a reason for the inclusion of nano-NiFe instead of macro-NiFe powder? As the authors pointed out, the use of macro-NiFe powder would make the comparison more valid.

Answer: Thank you for the suggestion. For this experiment, we wanted to underline the effect of each metal individually, independent of particle size (the latter mostly changing the overall yield, but not the product spectrum).

The NiFe powder used here was synthesized by collaboration partners to ensure an optimal mixture of the two metals (which cannot be ensured with commercial powders). The micropowder was commercially obtained so we could increase the metal to cofactor ratio as much as needed. Overall do these experiments really show, that the product distribution is due to the metal, not the particle size.

Minor comments:

1. For Figure 4A (Page 8), in order of decreasing Ni, the pie chart for NiFe₃ should be in the middle.

Answer: Thank you for the suggestion. The figure has been corrected accordingly.

2. On page 5, replace “In additional to” with “Additionally” or “In addition to”.

Answer: Thank you. The text has been corrected to “In addition to”.

3. For Figure 3A, the way the NMR spectra are stacked leads to a lot of overlap, which makes it difficult to interpret. The reviewer suggests that the NMR spectra be stacked in a way that avoids significant lap. Also, some of the text in Figure 3B is quite small and difficult to read.

Answer: We adapted Figure 3 in general and hope all is easier to read now.

4. In Table S13 (page 40) of the SI, the “0” in $\mu\text{Ni}0$ should be superscripted.

Answer: Thank you for the correction. The 0 has been superscripted.

We thank the reviewer for the kind and detailed revision of our paper!

Reviewer #3 (Remarks to the Author):

This paper deals with an important question in our understanding of co-factor evolution: why do so many cofactors contain an adenosine moiety? It's a bit of a perennial question which has not been definitively addressed. The current paper provides a fresh and new view: by comparing the reduction of nicotinamide adenine dinucleotide (NAD) (the molecule used as a cofactor in many enzymes) to a truncated nicotinamide molecule which lacks the adinonine, the authors arrive at a novel hypothesis where the latter might be imagined as being involved in mineral binding and the reduction process itself. I find this to be an intriguing proposal, and the data provide can be used to make such an argument, though remaining somewhat speculative.

Answer: We thank the reviewer for this kind assessment.

Below I provide major and minor comments interspersed. "investigate the ability of naturally occurring Ni- and Fe-containing allows to replace" → "alloys"

Answer: Thank you. The text has been corrected.

"nano-casting method by using tea leaves". Is there a more prebiotically relevant way to make such nanoparticles? Relying on plants for prebiotic chemistry is a little distant.

Answer: The overall process (reducing nanoparticulate oxides with H₂) is closer to serpenizing systems than one would think. For example, in these locations, Fe⁰ and Ni⁰ can be found underwater (*Chamberlain JA et al. Native metals in the Muskox intrusion. Can J Earth Sci. 1965 Jun 1;2(3):188–215; Kepezhinskas PK, et al. Native metals and intermetallic compounds in subduction-related ultramafic rocks from the Stanovoy mobile belt (Russian Far East): Implications for redox heterogeneity in subduction zones. Ore Geol Rev. 2020 Dec;127:103800.*), as a result of H₂-induced regeneration of the metal oxide.

The tea leaves only serve as a matrix (similar to what nano- or microporous rock systems would provide). Nevertheless, prebiotic synthesis of the minerals was not our main goal, rather a step towards obtaining minerals that would otherwise be available naturally in a prebiotic scenario. Producing our own minerals enabled us to guarantee that particle size, oxidation level, purity etc. are comparable for these reactions.

Figure 2: Please specify what ND is. Is it Not detected, or no data for example?

Answer: Correction done (explaining that ND means not detected in this case). Thank you.

Page 4: "or by direct electron transfer to NAD" Is there any evidence of a metal directly reducing a nicotinamide? If so what is the intermediate prior to addition of a proton? If there's no evidence, it's best to remove this.

Answer: Because we do not know the exact mechanism, we felt it was best to include all options during discussion. In nature, NAD is often reduced before being protonated (*K. Jungermann, R.K. Thauer, G. Leimenstoll, K. Decker, Function of reduced pyridine nucleotide-ferredoxin*

oxidoreductases in saccharolytic Clostridia, Biochimica et Biophysica Acta (BBA) - Bioenergetics, Volume 305, Issue 2, 1973, Pages 268-280).

With electrodes, NAD can also be reduced independently of the protonation state often leaving to the accumulation of dimers or 1,6-NADH (Ali I, Omanovic S. *Kinetics of electrochemical reduction of NAD⁺ on a glassy carbon electrode. Int J Electrochem Sci. 2013 Mar;8(3):4283–304*), instead of 1,4-NADH. We can remove the part if needed, but we actually think that it contributes positively to the discussion.

At the top of page 4 there is an interesting discussion comparing data presented in figure 2 but the discussion is quite vague and not definitive. Can the authors do more here? Or if not perhaps simply highlighting what is known and not known. Arrangement of the metal atoms, and not simply the number is likely important as well.

Answer: The first paragraph of page 4 was edited to include more details of the mentioned reaction that we hope makes the reading clearer. Considering our knowledge of the samples, and how difficult it is to analyze heterogeneous catalysis and amorphous surfaces, we cannot discuss much more at the current time. Also, considering that the metals were all prepared with the same protocol and leaf template, we expect that the arrangement of the metal atoms is not significantly different between each metal, which is supported by our STEM measurements (Supplementary Fig. S5 and S6).

We changed the text from

“After 4 h under H₂, 1,4- and 1,6-NADH formed in all samples at different yields, the reaction with nanoparticular NiFe₃ (nNiFe₃) yielding the most NADH (Figure 2). Control experiments starting from 1,4-NADH showed that 1,6-NADH is very likely a product of rearrangement from 1,4-NADH – with only a marginal influence of the used metals (*Supplementary Scheme S2, Tables S4 and S5, Figure S3*).”

to:

“After 4 h under H₂, 1,4- and 1,6-NADH formed in all samples at different yields, and the reaction with nanoparticular NiFe₃ (nNiFe₃) yielding the most of both molecules (Figure 2). Control experiments starting from 100% 1,4-NADH, showed that 1,6-NADH is a product of rearrangement from 1,4-NADH that occurs spontaneously without the need of a catalyst (*Supplementary Scheme S2, Table S4, Figure S3*). Starting from NAD⁺, the proportion of 1,6-NADH is higher in samples with higher NADH yields (NiFe alloys), showing that the accumulation of 1,6-NADH is not entirely independent of the metal (*Supplementary Table S5*).”

page 5: “Without metals, NMN does not react and remained stable (Supplementary Figures S8 and S9). Under Ar, NMN still got reduced due to the abundant iron in the mineral compound, but more slowly than under H₂ (Supplementary Tables S6 and S7).”
This is a confusing 2 sentences. Please revise for clarity.

Answer: Thank you. It now reads:

“Under Ar, NMN got reduced due to the iron in the mineral compound (NiFe₃) working as an electron donor, but more slowly than under H₂, where Ni can function as a hydrogenation catalyst (Supplementary Tables S6 and S7). Without metals, NMN does not react and remains stable, regardless of the gas phase (Supplementary Figures S8 and S9).”

Around page 5 I began to ask why is the first section of the paper with figure 2 present in the paper at all? This is a bit of a repeat of previous work - although valuable would it be better to place this in the supplemental and go straight to comparison of different molecule's reduction tendencies?

Answer: We can follow this sentiment. However, the higher efficiency of NiFe alloys in comparison to the lesser available reduced metals (Fe⁰ and Ni⁰) is a new finding that we find exciting, but also sets the stage for the NMN and NAD comparison: the reduction of NMN with NiFe alloys, Ni⁰ and Fe⁰.

Page 5 “The remaining 17% can in part be attributed to nicotinamide formation” What does this mean?

Answer: We have replaced “nicotinamide formation” by “degradation to nicotinamide” for clarification.

page 5: “and a variety of products was observed in” → “were”

Answer: Thank you, this was changed.

page 5: “the concentration of twice reduced 2c increased with once reduced 1,4-NMNH decreasing” please revise this was not clear.

Answer: We changed the sentence to: “the concentration of twice reduced 2c increased steadily overtime and correlates to 1,4-NMNH's concentration decreasing.”

page 5: “Under Ar, 2d did not form at all, “ was this or without metal? and with which one if there was a metal? This is a recurring confusion in the paper. Can the authors consider a naming scheme to deal with this. For example Ar(NiFe) or something such that the presence of a metal is more clear?

Answer: We thought about adding a scheme as proposed, but then realized it over-complicates the reading. We always indicated the no metal reactions as such and in all “under Ar” referring sentences there is now a clear association to the metal. Also in our SI our controls are clearly indicated and listed and referred to.

page 6: “total amount of reduced NMN remained relatively stable”. It might be helpful to add a line showing this in the plot.

Answer: We assume the reviewer is referring to Figure 3D. We applied the correction as suggested. We also edited the text to be more specific about how stable it is, which we hope addresses the reviewer's concerns. Where it said “total amount of reduced NMN remained

relatively stable, exceeding 60%.”, it now reads: “total amount of reduced NMN remained relatively stable, between 67 and 69%.”

Figure 3B. Is there any evidence for the 6e- reduction of of NMN without an intermediate? if not, it is best to remove this. The current data seems to lack the required time resolution to show this.

We use the following argument for this: “While the fully reduced species 2d formed quickly and its concentration stagnated, the increasing concentration of twice reduced 2c correlates to 1,4-NMNH decreasing. This indicates that not all reductions are a step-wise process (s. Figure 3B), especially in the case of 2d.”

The speed at which 1,2,4,6-NMNH₅ accumulates does not seem to increase with increasing 1,4,6-NMNH₃ concentration, which would be the intuitive intermediary. While we do not know exactly what the system is, we believe it is different than that of 1,4,6-NMNH₃, and suggested our best prediction. As we cannot be 100% certain of the exact overreduction route, we are now indicating the three possible ones in Fig. 3B with dashed arrows.

Figure figure legend: Full arrows represent proposed reaction mechanisms” Perhaps mechanism implies an understanding of how the electrons and atoms are moving - maybe delete it and simply call them ‘proposed reactions’ ?

Answer: We change this to “proposed reduction patterns”.

page 7: “As both nNi and nFe visibly” please introduce the “n” abbreviation as nomenclature earlier in the paper, perhaps near the tea leaves casting method. Are the data associated with this section comparable to figure 2 in having used the “n” form? Please label consistently.

Answer: In page 4, under Figure 2 we introduce “n” for nanoparticular, in the case of nNiFe₃ alloys, which we then applied to different metals throughout the text. At the start of page 8 “μ” is introduced for micropowder. We now are introducing nanoparticular equals “n” also for Ni and Fe in the mentioned sentence – we are not sure, however, if this now introduces some redundancy.

Figure 4: what was the amount of time? Are there time course data?

Answer: We thank the reviewer for their attention. The time has been specified to be 4h in the legend of the figure.

page 8: “The metal-cofactor ratio was 200:1 to guarantee the detection” This change makes it difficult tom compare to anything which was above (a 1:1 ratio). I am always loath to propose new experiments but this is an important part of the paper and the variation in the ratios used in the experiments makes the reading and conclusions fragmented. If possible please investigate the ratio dependence.

Answer: In this experiment we looked to highlight the differences between Ni and Fe. 200:1 helps us achieve enough reduction of NMN with micropowder that would otherwise be achieved with much less nanopowder in the same amount of time and the same experimental conditions. With

1:1 micropowder, not enough NMN would be reduced in 4h, in order to detect it. We experimented with 50:1 micropowder, and got the same products, suggesting that this is a reproducible result (Fig. S32 and Table S18).

Page 8: “This could also explain the fast formation of its fully reduced product 2d shown in Figure 4C. NAD in a staggered formation could only absorb partly on the surface, avoiding overreduction” But the NiFe₃ seems to make 1-4NADH quite well?

Answer: We would like to clarify that at the start of the paragraph it is written: “Nickel has long been recognized as a hydrogenation catalyst⁵¹, – but why does it, when not combined with Fe, only reluctantly reduce NAD (Figure 2 and ¹²) and yet overreduce NMN”. Simply put: why does Ni⁰ work so differently to hydrogenate NAD than it does to NMN?

In the context of our work, NAD is interesting because heterogeneous hydrogenation always leads to 1,4-NADH being the main product, even with pure nickel, and not to overreduction or hydrolysis. NiFe₃ does lead to higher and significant yields, but it never overreduces NAD to, for example, 1,2,4,6-NADH₅. The point being discussed is not about the rate of reduction, but rather be effectiveness of it. We tried to clarify:

“This could also explain the fast formation of its fully reduced product 2d shown in Figure 4C. NAD in a staggered formation could only absorb partly on the surface, avoiding *such overreduction reactions*”.

Paragraph at bottom of page 8 and going into page 9: This is interesting but not discussed in a substantial way: why is this? If the authors don't want to elaborate it could be cut.

Answer: The discussion continued from that point until the end of the current chapter. We have made all the paragraphs a single one in order to connect the topics more clearly.

page 9: “ seems to have a dampening effect on NMN (over)reduction when both cofactors are “ This was an important experiment and provides an interesting result. It's related to the above brief discussion of why Ni and Fe seem to reduce differently. It's notable that the amount of 1,4NADH is lowered in the mixed experiment compared to the unmixed (13.6% vs 19.8%). Perhaps there is some contribution from the metal and the organic.

Answer: Thank you for the nice comment. It is something we noticed and are planning to explore this in detailed future work. Because of the significant impact on overreduction, we felt it was significant to add to the paper, but further work is necessary to understand the complete effects.

in table 1: the individual co-factor concentrations were both 12mM if I understand it, perhaps that can be indicated more clearly in the table which currently simply says 12mM.

Answer: The legend of the figure as well as the table were improved. The legend now says:

“Overview of yields of mixtures of NAD⁺ and NMN in comparison to separate reduction with H₂ gas. The left column shows the quantification of 12 mM of NAD⁺ (n=3) and 12 mM of NMN

(n=3) in individual reactions with μNi and μFe . The right column shows a reaction mixture of the same amount of NAD^+ and NMN (12 mM each) (for all n=3).”

Also in the table: is a two-tailed comparison the correct one here? I'd check back to multiple comparisons - Bonferoni corrections / anova here.

Answer: Thank you for this comment. We double-checked what method is appropriate here and we have to stick with the two-tailed comparison, as it is the more conservative one (so we can trust the resulting significance more) opposed to one-tailed. Anova and Bonferoni are both for larger data sets than what we have. Bonferoni adjusts significance when performing multiple statistical tests to check for false positives. In our case (n=3) this would not lower the significance in a significant way, it is a bit “too much” for our small data set. We hope this is understandable.

page 9: “As the second signal has a lower oxidation potential” we almost always write “reduction potential” because the directionality of the reaction (what is substrate, and what is product) enters into the Nernst equation. Please revise the paper to discuss reduction potentials, and when referring to the c.v.'s the use of “cathodic” and “anodic” currents can be used to indicate current detected at negative becoming or positive becoming potentials. This made the text difficult to interpret, for example “Cyclic voltammetry experiments showed that the oxidation potential of single-reduced species, while all further side (overreduction) products fall behind. Concerning the single-reduced side product 1,6-NADH, we assume it to have a comparable redox potential as 1,4-NADH, although” Please consider a way to write this using reduction potential, cathodic, and anodic.

Answer: Thank you, we adapted the text in both manuscript and SI according to the suggestions.

The performance of the electrochemistry experiments is really great, but the potentials reported seem completely off with our knowledge:

For example the Cyclic voltammetry starting at S30. The reduction potential of NAD^+ is $\sim -580\text{mV}$ vs SCE, but the CV shows something very different with the cathodic current appearing at about -1V vs SCE. This is very negative: what is it? It is negative enough that it could be H^+ reduction at the electrode. in S30B, the anodic current appears at $+500\text{mV}$ vs SCE. This is very positive and more coincident with reactions involving oxides/ O_2 . In table S12 the listed “oxidation potentials” (which I presume are the potentials associated with the major cathodic currents in the c.v.) are also in this regime, which is outside of the redox space of nicotinamides.

Answer: To our understanding, for most non-optimized CVs, the reduction of NAD^+ to 1,4-NADH is very slow and inefficient. It starts at -500mV , as stated by the reviewer, but in very low amounts. The fast reduction of NAD, leads to the observed peak we see at -1.1V , which is mostly producing a mixture of NAD_2 and other species like 1,6-NADH as a product of further reduction of the dimer (Meyer J, Romero M, et al. *Experimental insights into electrocatalytic $[\text{Cp}^*\text{Rh}(\text{bpy})\text{Cl}]^+$ mediated NADH regeneration.* *Sci Rep.* 2023 Dec 16;13(1):22394).

We have removed the SI Figures starting from NAD^+ and kept the ones with 1,4-NADH, as it is more relevant to (relatively) compare to the products of our experiments.

(back to the main text)

page 12: “The immediate presence of 2d” as with an above comment, the time course data does not seem at a small enough increment to show this.

Answer: We changed “immediate” to “early” and discuss the association of NMN to a metal surface in the following paragraph as well, supported by existing literature.

in S14, can the authors provide the expected position of the methine carbon they are referring too and a picture of that dimer molecule to aid in interpretation? This figure lacks reference to in the SI as well.

Answer: The requested change has been applied, now in the Supplementary Data file as Fig. SD5, in addition to similar changed to figures SD4, SD7 and SD8 and SD9.

Final comment: The authors used phosphate buffer in their experiments. Metals and phosphates interact strongly, and the interaction between Fe and Ni is different. Thus one could make hypotheses that the differences observed in the reductive tendencies is due to a blanket of phosphate on the minerals. I strongly encourage the authors to perform experiments where this can be investigated prior to publication. One might consider carbonate buffer due to its relevance in natural systems, comparing reduction products from NAD vs NMN with the metal alloys used and also pure Ni and Fe. Without some isolation of the effect of phosphate we are left wondering about its importance.

Answer: Thank you for the attentive comment. We have repeated the experiments using Fe powder with carbonate instead of phosphate buffer as suggested – the interaction of Fe and phosphate has been established previously in the manuscript by STEM measurements. Results show a similar reduction pattern with both buffers (Supplementary Tables S15 and S16 or Tables 1 and 2 above), when using μFe . We hypothesize that the accumulation of iron precipitates is rather a consequence than the cause of the reactions taking place. Because Fe is a better reductant than Ni (which is a better hydrogenation catalyst), it oxidizes faster, reducing water or NAD^+ in the process. Fe^{2+} then precipitates with the buffers’ anions. It is of course very possible that the layer of Fe salts hinders further reduction or changes the type of interaction with NAD/NMN – we were discussing this option previously but expanded this a bit now.

The discussion and conclusion now include the carbonate findings, including the following statements in the SI:

“The reduction of NAD^+ with Fe^0 in carbonate buffer led to a wider variety of uncharacterizable side products (Fig. S32) than during reactions in phosphate buffer (**Error! Reference source not found.**). The peaks of the side products are less defined than the peak of 1,6-NADH which appears in the same region of the spectra. This effect was also visible in controls without Fe^0 , although in even lesser amounts. The amount of each of the previously characterized main products (1,4-NADH, 1,4-NMNH and NMNHOH), however, is affected.”... “Extra peaks were also found in the NMN carbonate samples, as they were in the samples with NAD^+ , leading to a

significant loss of the starting material (Table 16; Table 2 above), but without affecting the main products yield. In conclusion, phosphate is seemingly less reactive and thus a more reliable buffer.”

We thank this reviewer and all reviewers whole-heartedly for their comments and suggestions.
We hope our changes and answers resolve any concerns and add a better perspective.

Answers to the reviewers

Reviewer #2 (Remarks to the Author):

The reviewer thanks the authors for taking the time and effort to respond to the comments regarding the NMR assignments and folding dynamics. The authors have now provided a suite of new NMR and computational data to back up their arguments. With respect to the new computational data, the reviewer has no further questions or comments. This study is very well done and backs up their hypothesis.

Thank you! Much appreciated.

When it comes to the NMR data, the reviewer still has some comments/suggestions. With respect to the two NMR resonances assigned to the C2 proton of 1,4,6-NMNH₃, the authors now suggest these belong to two stereoisomers (more specifically, conformers) brought about by rotation of the amide. This assignment could very well be correct, but the reviewer is still skeptical. Restricted rotation about this C-C bond to the amide might indeed be brought on by conjugation with the double bond, but why then would similar conformational isomerism not be observed with 1,4-NMNH and 1,4-NMNH₂OH? The authors have already carried out quite a large number of NMR experiments, and the reviewer isn't asking for more. The only way to really be sure about these assignments is to make a pure standard of 1,4,6-NMNH₃ (as well as the other compounds), but this may be outside the scope of the present study. The reviewer suggests that the authors only propose that two stable conformational isomers of 1,4,6-NMNH₃ exists, and not be so explicit about what each one is. For example, isomers involving two conformations of the piperidine ring seem possible as well, similar to those seen for substituted cyclohexene rings (e.g., see <https://pubs.acs.org/doi/pdf/10.1021/cr60308a004>). Is there similar literature the authors can cite for piperidine systems? The reviewer is not trying to be overly pedantic, but the NMR spectra are quite complicated, while the proposed products may also be conformationally dynamic giving rise to further spectral complexity. It would be incredibly easy to make a misassignment, and the reviewer, for the authors' own sakes, wants to make sure all the possibilities have been thoroughly considered.

Upon further reflection, we agree with the reviewer that the assignment is unnecessarily specific, as being more vague does not affect the broader scope of this manuscript. We made a few changes to the manuscript that we hope reflect a better understanding of the system and its limitations. We repeated some of our reactions to confirm the assignments we did were correct. These repetitions (see Fig. 1 and 2 below) confirm that we are looking in both cases at nicotinamide rings that are twice reduced at the previously determined positions: 1,4,6. As we cannot assign the rest of the molecule with certainty beyond this, so we decided to change the labelling of these peaks from "1,4,6-NMNH₃", to "1,4,6-products", indicating that these products show the same overreduction pattern – but also showing that we cannot assign the rest of the molecule beyond any doubt. We also removed the SI Schemes suggesting specific conformers to avoid misinterpretation. We think this procedure aligns perfectly with what the reviewer suggested while conserving the main conclusion of our manuscript.

Fig. 1: HSQC of a 4 h reaction of NMN with Ni (50:1 ratio to cofactor) and H₂.

Fig. 2: TOCSY of a 4 h reaction of NMN with Ni (50:1 ratio to cofactor) and H₂.

The text in the maintext reads now as follows:

“The two peaks assigned 2c have been attributed to twice reduced nicotinamide rings as depicted in Figure 3B. This is either due to conformational isomers or possibly other deviations of the molecule apart from the nicotinamide ring.”

We reflect how the complexity of the mixture hinders us to assign everything in the legend of Fig. 3:

“Due to the complexity of the mixture not all peaks could be assigned.”

Finally, we also found some NMR literature on similar systems that helped us with understanding the peaks better. We are now citing them in the Supplementary Data:

1) Barret et al.: Photoswitchable Hydride Transfer from Iridium to 1-Methylnicotinamide Rationalized by Thermochemical Cycles (10.1021/ja508762g)

2) Acheson and Paglietti: Reduction of Some I -Substituted Pyridinium Salts (10.1039/P19760000045)

In the case of NMNH₂OH, the authors argue that hydration will result in two diastereomers, cis and trans with respect to the ribose ring. This makes sense, but why then does one diastereomer seem to decrease as the reaction goes on longer? Since the authors have 1,4-NMNH available as a standard, what happens when they dissolve it in water under the experimental conditions except without hydrogen gas? Does it readily undergo hydration? Are two diastereomers observed? This is a simple enough experiment to do that will support these NMR assignments.

We performed these experiments with and without metal, showing that NMNH₂OH is a direct product of 1,4-NMNH hydrolysis. We took these results to polish a bit our understanding of what happens with different metals as follows in the maintext:

*“...two main products: 1,4-NMNH and **2b**, the latter being the hydrolysis product of the former, which forms with and without a metal catalyst (**Supplementary Scheme S7, Table S17, and Figure S31**).”*

*“The lacking overreduction with Fe also explains the accumulation of the hydrolysis product **2b** in Fe-only reactions: if more 1,4-NMNH can be formed without being further reduced, the more of it can be hydrolysed to **2b**. So with a strong hydrogenation catalyst such as Ni, overreduction likely prevents hydrolysis.”*

We were however not able to reproduce the double peak in any of our repetitions, possibly because we could not get hands on exactly the previously used alloy powder (NiFe₃). So we decided to use a deconvolution tool to take out the unassigned peak's area for quantification of NMNH₂OH over time. Over time, the area of this peak decreases, ultimately contributing only 1% to the overall quantification at 3h. Because of the neglectable contribution and the fact that we do not see the peak in any of the 4h experiments (regardless of metal used), we did not take this peak into account for any of the 4h experiments. We write about this in the maintext:

“The hydration product NMNH₂OH (2b) was matched to a peak at 7.34 ppm in the ¹H-NMR (Supplementary Data Figures SD9 and S10), the adjacent smaller, overlapping peak at 7.35 ppm could not be assigned beyond a doubt area of the smaller peak was thus subtracted with a deconvolution tool (Supplementary Table S10–12).”

We also added more explanation on the deconvolution in the legend for Supplementary Table S10 in which we show the data for this measurements:

“The reactions that ran for 1, 2 and 3 h had a smaller, overlapping peak at 7.35 ppm, adjacent to NMNH₂OH’s peak. This could not be assigned beyond a doubt, so the peak’s area was subtracted by using a deconvolution tool (Mestrenova v.15.0.1). The contribution to NMNH₂OH’s peak area reaches 1% at the 3h measurement. In all 4h experiments in this paper we could not observe this additional peak and did thus not account for its presence.”

Furthermore did we remove the SI Scheme suggesting specific conformers for NMNH₂OH to avoid misinterpretation.

Some additional considerations when it comes to stereochemistry are with compound 2d. Reduction should lead two diastereomers of 2d as well, with the amide being either cis or trans with respect to the ribose ring. Is there any evidence of two diastereomers in the NMR spectra?

This is an excellent point and the reason why we decided to name 2d also 1,2,4,6-product instead of 1,2,4,6-NMNH₅. This does not change the main message of the paper, but accounts for the fact that we might not see other products with the same reduction pattern in the nicotinamide ring.

At the end of the day, since the main point of the manuscript is that NAD⁺ is only reduced to NADH, while NMN is completely reduced, the NMR data definitely support this main conclusion. The reviewer thinks publication is still warranted as long as the authors include a little more discussion of the NMR data for the NMN experiments in the main text. Specifically, they should have at least one paragraph dedicated to how assignments were made, and acknowledge that given the complexity of the spectra, some assignments are tentative.

We thank the reviewer and incorporated more forwardness on the spectra complexity in the maintext by renaming our twice and thrice reduced products e.g. “1,4,6-products” instead of stating they are definitely 1,4,6-NMNH₃.

Reviewer #3 (Remarks to the Author): The authors have improved the manuscript in the review process. One final suggestion is to include the Meyer paper <https://doi.org/10.1038/s41598-023-49021-4> as a citation and use it to mention the reduction potentials; this would be helpful for readers.

We thank the reviewer for their previous comments that helped improving the paper. We added the sentence

“Previous literature on NAD cyclic voltammetry was consulted for the CV interpretation”

with the according reference where we present the CV data in the Supplementary information (SI p. 44).